# Attosecond formation of charge-transfer-to-solvent states of aqueous ions probed using the core-hole-clock technique

E. Muchová [1] ✉, G. Gopakumar[2], I. Unger[2,3], G. Öhrwall [4], D. Céolin[5], F. Trinter [6], I. Wilkinson [7], E. Chatzigeorgiou[2], P. Slavíček [1], U. Hergenhahn [6], B. Winter [6], C. Caleman [2,3] & O. Björneholm[2] ✉

Charge transfer between molecules lies at the heart of many chemical processes. Here, we focus on the ultrafast electron dynamics associated with the formation of charge-transfer-to-solvent (CTTS) states following X-ray absorption in aqueous solutions of $Na^+$, $Mg^{2+}$, and $Al^{3+}$ ions. To explore the formation of such states in the aqueous phase, liquid-jet photoemission spectroscopy is employed. Using the core-hole-clock method, based on Auger−Meitner (AM) decay upon $1s$ excitation or ionization of the respective ions, upper limits are estimated for the metal-atom electron delocalization times to the neighboring water molecules. These delocalization processes represent the first steps in the formation of hydrated electrons, which are determined to take place on a timescale ranging from several hundred attoseconds (as) below the $1s$ ionization threshold to only 20 as far above the $1s$ ionization threshold. The decrease in the delocalization times as a function of the photon energy is continuous. This indicates that the excited electrons remain in the vicinity of the studied ions even above the ionization threshold, i.e., metal-ion electronic resonances associated with the CTTS state manifolds are formed. The three studied isoelectronic ions exhibit quantitative differences in their electron energetics and delocalization times, which are linked to the character of the respective excited states.

Charge-transfer processes between atoms and molecules lie at the heart of many fundamental processes, including photosynthesis. Electron-transfer processes in an aqueous environment are of particular importance[1]. These reactions start with photoabsorption and threshold electron-ejection processes, leading to the population of charge-transfer-to-solvent (CTTS) states. These states consist of a hydrated parent species with an excited electron, stabilized by solvent polarization[2,3]. In water, CTTS states are short-lived, and the excited

electron rapidly becomes solvated or, specifically in water, hydrated. The dynamics of the CTTS states of valence-excited hydrated anions have been extensively studied on femtosecond (fs) to nanosecond (ns) timescales using time-resolved spectroscopy[4–9] or, more recently, molecular dynamics (MD) simulations[10,11]. The picture that emerges from these studies is a sequential ultrafast process, starting from initial photoexcitation and leading to the formation of hydrated electrons. First, a locally excited or "trapped" state with a lifetime of 50–100 fs is

[1]Department of Physical Chemistry, University of Chemistry and Technology, Prague, Technická 5, 166 28 Prague, Czech Republic. [2]Department of Physics and Astronomy, Uppsala University, Box 516, SE-751 20 Uppsala, Sweden. [3]Center for Free-Electron Laser Science, DESY, Notkestr. 85, 22607 Hamburg, Germany. [4]MAX IV Laboratory, Lund University, Box 118, SE-22100 Lund, Sweden. [5]Synchrotron SOLEIL, L'Orme des Merisiers, Saint-Aubin, BP 48 91192 Gif-sur-Yvette Cedex Paris, France. [6]Fritz-Haber-Institut der Max-Planck-Gesellschaft, Faradayweg 4-6, 14195 Berlin, Germany. [7]Institute for Electronic Structure Dynamics, Helmholtz-Zentrum Berlin für Materialien und Energie, 14109 Berlin, Germany. ✉e-mail: muchovae@vscht.cz; olle.bjorneholm@physics.uu.se

formed. This first state subsequently relaxes to a more delocalized, second state, described as a "wet" or "pre-equilibrated" but not yet fully hydrated electron. On a 100–400 fs timescale, solvent reorganization converts this second state into the equilibrated ground state of the hydrated electron[7–10,12]. Despite extensive study, it has, however, not been possible to characterize the initial valence-excitation mechanism and associated electron delocalization process due to their exceptionally short timescales. As we demonstrate here, this is, however, possible for core-level excitations, enabling quantification of the excitation energy dependence, energetics, and upper-limit attosecond timescales of the initial steps in the electron delocalization for three exemplary, isoelectronic, monoatomic ions in aqueous solution.

Recent advances in attosecond science bring the inherent timescales of electron dynamics into focus[13–15]. A direct way of probing attosecond dynamics is to use attosecond-duration photon pulses. The implementation of extreme ultraviolet (EUV) attosecond pulse trains and isolated attosecond pulses, typically produced by high harmonic generation, has enabled ultrafast electron dynamics—such as time delays in photoemission and charge migration after prompt ionization of gases, liquids, and solids—to be probed directly, as recently reviewed in ref. 16. Attosecond soft X-ray pulses produced at free-electron lasers have also recently been used to probe, e.g., electronic wave-packet dynamics associated with Auger–Meitner (AM) decay in NO molecules[17] and valence ionization of liquid water in the first attosecond-duration X-ray pump, attosecond-duration X-ray probe study on a condensed-phase system[18].

Attosecond dynamics have also been probed using indirect methods, relying on some other dynamic process to act as an internal clock. For example, dissociative nuclear dynamics in $H_2$ has been used to track sub-femtosecond autoionization dynamics[19], and molecular orientation with respect to a photon beam has been used to resolve the zeptosecond time delay between electron emission from the two centers of the hydrogen molecule[20]. In $CO$[21] and $NO$[22] molecules, the attosecond Wigner time delay for photoionization has been reconstructed from the photoelectron angular emission distribution in the molecular frame. The core-hole-clock technique belongs to the same group of indirect methods of probing attosecond electron dynamics; here, the lifetime of a core hole, typically produced via photo-nionization or photoexcitation, is used as an internal temporal reference for ultrafast phenomena that occur in parallel with core-hole relaxation[23–26].

In the case of electron-transfer processes, the core-hole-clock technique is based on the identification of separate contributions to AM electron spectra[27] (see Fig. 1) and the derivation of dynamic timescales from energy-resolved experiments. The AM spectrum can be compared in some ways to a photograph taken with a long exposure time, where motion can be observed even though the moving object is too fast to be sharply imaged. AM decay is an exponential process described by a time profile $e^{-t/\tau_{1s}}$, where $\tau_{1s}$ is the lifetime of the $1s$ core hole and $t$ is the time since $1s$ electron excitation. If no other process competes with the AM decay, the AM spectrum contains only spectator lines in the energetic region where normal AM signals are expected (the related participator lines occur at higher kinetic energies), see the upper part of panel (b) in Fig. 1. If charge transfer occurs on the same timescale as AM decay, the AM spectrum contains contributions from both normal and spectator AM decay; see the middle section of panel (b) in Fig. 1.

Previous core-hole-clock studies have experimentally revealed charge-transfer times ranging from 9 fs for weakly coupled $N_2$ physisorbed on graphite[23], over 320 as for S chemisorbed on ruthenium[26], and down to tens of attoseconds for Xe adsorbed on metal surfaces[28]. In the context of liquid water and aqueous solutions, processes such as

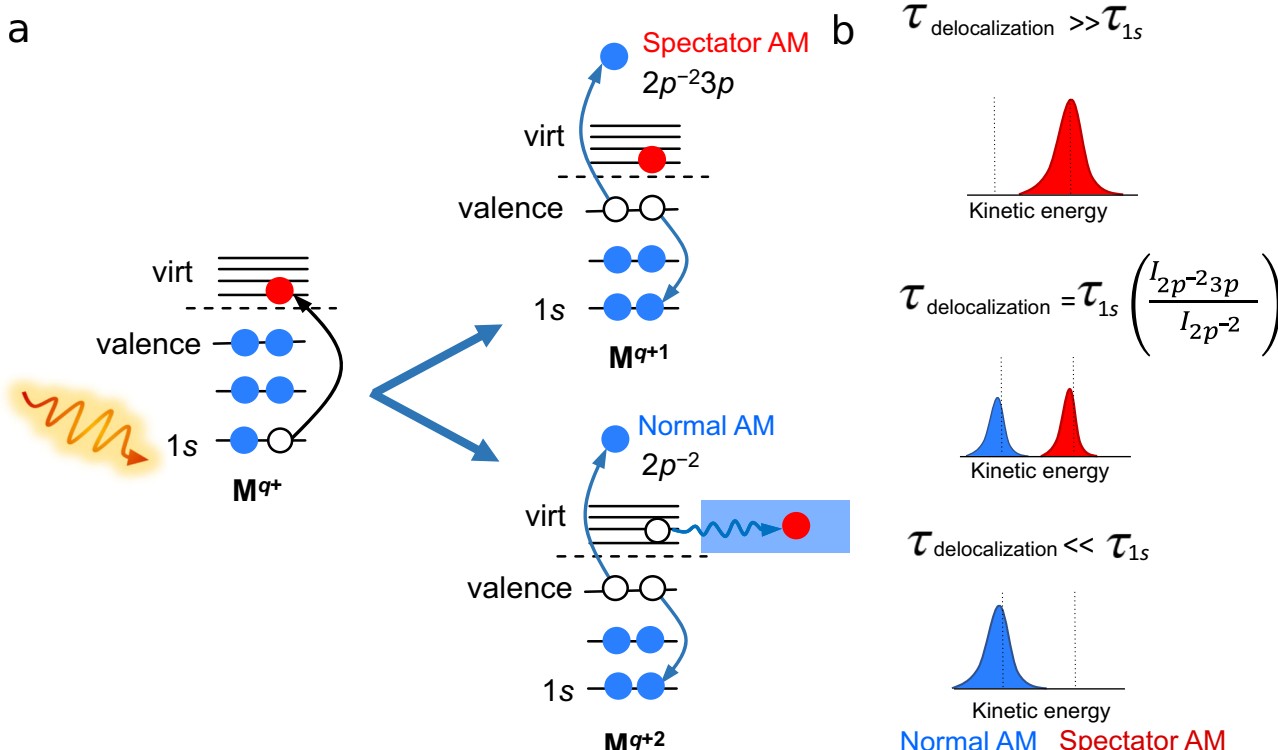

**Fig. 1 | Schematic representation of the core-hole-clock method. a** In the first step, below the metal-cation ionization threshold, a core electron is excited to the virtual orbitals and leaves a hole in the coreshell. If the excited electron remains localized in the atomic orbital, a spectator Auger–Meitner (AM) electron is emitted (upper process, yielding the red peak in panel **b**). If the electron delocalizes into the environment, a normal AM electron is emitted (lower process, yielding the blue peak in panel **b**). **b** Depending on the core-hole lifetime, $\tau_{1s}$, and the delocalization time, $\tau_{delocalization}$, different spectator and normal AM electron signal-intensity ratios are obtained.

delocalization of electrons excited into CTTS states have been shown to occur on the timescale of $2p$ core-hole lifetimes for Cl$^-$ ions ($\approx 6$ fs)[29,30]. The core-hole-clock concept has also, for example, been used to show that the timescale for electron hopping from water, core-excited at the post-edge region into unoccupied delocalized orbitals, is <500 as[31], while cations significantly slow down this delocalization to the few-femtosecond timescale[32]. For hydrated cations, the core-hole-clock method has been used to determine a few tens to few hundreds of femtoseconds timescales of $2p$ core-level intermolecular Coulombic decay (ICD, a special type of non-local autoionization decay[33]) for Na$^+$, Mg$^{2+}$, and Al$^{3+}$[34] as well as K$^+$ and Ca$^{2+}$[35]. Based on line-shape analyses of core-level photoelectron spectra, even faster intermolecular Coster−Kronig-like processes have been reported for $2s$ core-level ICD in hydrated Na$^+$, Mg$^{2+}$, and Al$^{3+}$ ions, with inferred lifetimes of 3.1, 1.5, and 0.98 fs, respectively[36]. More recently, the corehole-clock method was used to quantify the delocalization dynamics following excitation or ionization of aqueous-phase K$^+$ and Cl$^-$[37]. Given the chemical selectivity afforded by core-level spectroscopy and the possibility to probe electron dynamics on few- and even sub-femtosecond timescales, the core-hole-clock technique is expected to remain a powerful tool for the investigation of even complex, biologically relevant systems. The potential of this method will correspondingly continue to be explored, despite emerging opportunities for time-resolved pump-probe experiments with attosecond X-ray pulses, and may well serve as a calibration method for the latter. Furthermore, currently, the application of attosecond X-ray pulses to aqueous-phase photoemission spectroscopy is hampered by the low pulse repetition rates of the available light sources, forcing the use of overly high pulse intensities and leading to significant and difficult-to-quantify space-charge effects.

In this paper, we investigate the first steps of charge separation associated with CTTS formation, which take place on the sub-femtosecond timescale. We do this by exploring the X-ray-induced electron dynamics that occur following excitation or ionization of the $1s$ electron of the hydrated Na$^+$, Mg$^{2+}$, and Al$^{3+}$ ions (all with isoelectronic neon-like $1s^2 2s^2 2p^6$ electronic configurations). With increasing ionic charge across this sample series, the ion−water distance decreases, and the interaction changes from pure ion−dipole to a combination of ion−dipole and dative covalent bonding[36]. By selectively exciting or ionizing the $1s$ electron from the solvated metal ions, we use the core-hole-clock method in a similar way to that described in ref. 37, determining the delocalization timescales of the initially localized excited-state electronic wave packets into states that are predominantly delocalized over the surrounding water molecules.

The experimental findings are accompanied by ab initio calculations that describe the character and energetics of the initial CTTS states and the final states of the AM decay. Comparison between the hydrated Na$^+$, Mg$^{2+}$, and Al$^{3+}$ ion results reveals how an increasing ionic charge, and consequently a more organized hydration shell, influences the time it takes for the excited electron to delocalize from the ion to the water environment. These results correspondingly provide insights into how the attosecond electron dynamics related to the first steps of charge separation change with increasing electronic coupling, specifically in going from a system dominated by ion−dipole bonding towards a system with appreciable coordination bonding.

## Results and discussions

### X-ray absorption

We start with a brief overview of the studied processes from a time-dependent perspective. Upon the interaction of a photon with a $1s$ electron, an excited-state electronic wave packet is formed. This wave packet is initially localized in the proximity of the excited atom and is of $p$ symmetry within the dipole approximation. For isolated atoms, the wave packet gradually transforms into one of the $np$ atomic states, if permitted by the energy resonance condition. The situation is,

however, different for the hydrated ions. The initial electronic wave packet scatters from neighboring water molecules, forming complicated wave packets, which gradually evolve into CTTS states. The system still resides in a continuum of doubly ionized and charge-separated, hydrated electron states, and the excited-state eigenstates are not fully formed during the finite lifetime of the $1s$ core-hole states. The absorption process is expected to take place on a sub-femtosecond timescale, which is close to the $1s$ core-hole lifetimes of the studied ions, as further discussed below. The non-stationary evolution of the electronic wave packet of the core-hole state is probed by the AM decay, which provides a snapshot of the degree of wave-packet (de)localization, thereby reflecting the electron dynamics.

Panel (a) of Fig. 2 shows the integrated KLL electron PEY-XAS (the spectra are based on the 2D maps presented below, where the integrated kinetic-energy ranges were 982.5–986.5 eV for Na$^+$, 1173–1178 eV for Mg$^{2+}$, and 1378–1382 eV for Al$^{3+}$), which approximate the true $1s$ XAS. Below the $1s$ ionization threshold, marked by dotted lines in Fig. 2, the X-ray absorption is dominated by electronic transitions into unoccupied orbitals. Photoabsorption is much more probable above the $1s$ ionization thresholds (1076.7, 1309.9, and 1567.7 eV for the hydrated Na$^+$, Mg$^{2+}$, and Al$^{3+}$ cations, respectively[36]). For an isolated atom, the main resonant transitions are $1s \rightarrow np$, $n \geq 3$, with the relative transition strengths decreasing with $n$. The unoccupied orbitals of the hydrated metal cations are, however, affected by the surrounding water, and the atomic notation is not fully adequate in the case of the CTTS states. In spite of this, in the text, we use the $np$ term to describe the virtual orbitals with a significant contribution from the metal-cation $np$ orbital.

We now focus on the character of the core-excited CTTS states from the perspective of electronic-structure theory. Simulated XAS are presented in Fig. 2 panels (b)–(d), which were performed at the SRC2-R2 level with a cc-pCVTZ basis set on the cation, a cc-pVTZ basis set on the water molecules, in a polarizable continuum. The SRC2-R2 spectra agree well with the results obtained at the CVS-EOM-EE-CCSD level and within the RT-TDDFT level, see Supplementary Fig. 3 in the SI. The spectra calculated for a randomly selected structure of the $[M(H_2O)_6]^{n+}$ cluster are shown in panel (b) of Fig. 2. The weight of the localized character is calculated as $(1-CT)$ and projected as a color bar onto the oscillator strength (gray bar). The respective unoccupied NTOs are shown as insets in Fig. 2 panel (b). As can be seen from the figure, the spectra below the $1s$ ionization threshold correspond to the transitions from the metal-cation $1s$ orbitals to the unoccupied orbitals with partially localized $np$ and partially delocalized character.

The character of the unoccupied orbitals differs for each cation. The Na$^+$ hydration shell is loose and allows the electron density to be accommodated in the space between the cation and the closest water molecules. The shape of the orbital resembles that of the $np$ state of an atom. This can be seen in most of the transitions, the local contributions (violet bars) to the excited states are relatively high for all excited states, see Fig. 2 panel (b). For Al$^{3+}$, the water hydration shell is tight and organized (a "cage" is formed by the surrounding water molecules), and the resulting confined space cannot fully accommodate the diffuse $np$ state in the same way as observed for Na$^+$. Consequently, the electron density is partially localized near the excited cation and largely scattered over the neighboring water molecules. The local contributions (orange bars) to the excited states are negligible below the $1s$ ionization thresholds and are approximately half of the total intensity above the ionization threshold, see Fig. 2 panel (b). The character of the excited states for the Mg$^{2+}$ cation is between those of the Na$^+$ and Al$^{3+}$ ions. A similar picture can be derived from the $\sigma_e$ parameter, the root-mean-square size of the excited electron with respect to the hole localized on the metal center, depicted in Supplementary Fig. 1 in the SI. In the case of the smaller clusters, for Na$^+$, a significant number of the excited states with a non-zero oscillator strength are very diffuse (high value of $\sigma_e$), while for Al$^{3+}$, the $\sigma_e$ value is much smaller, which

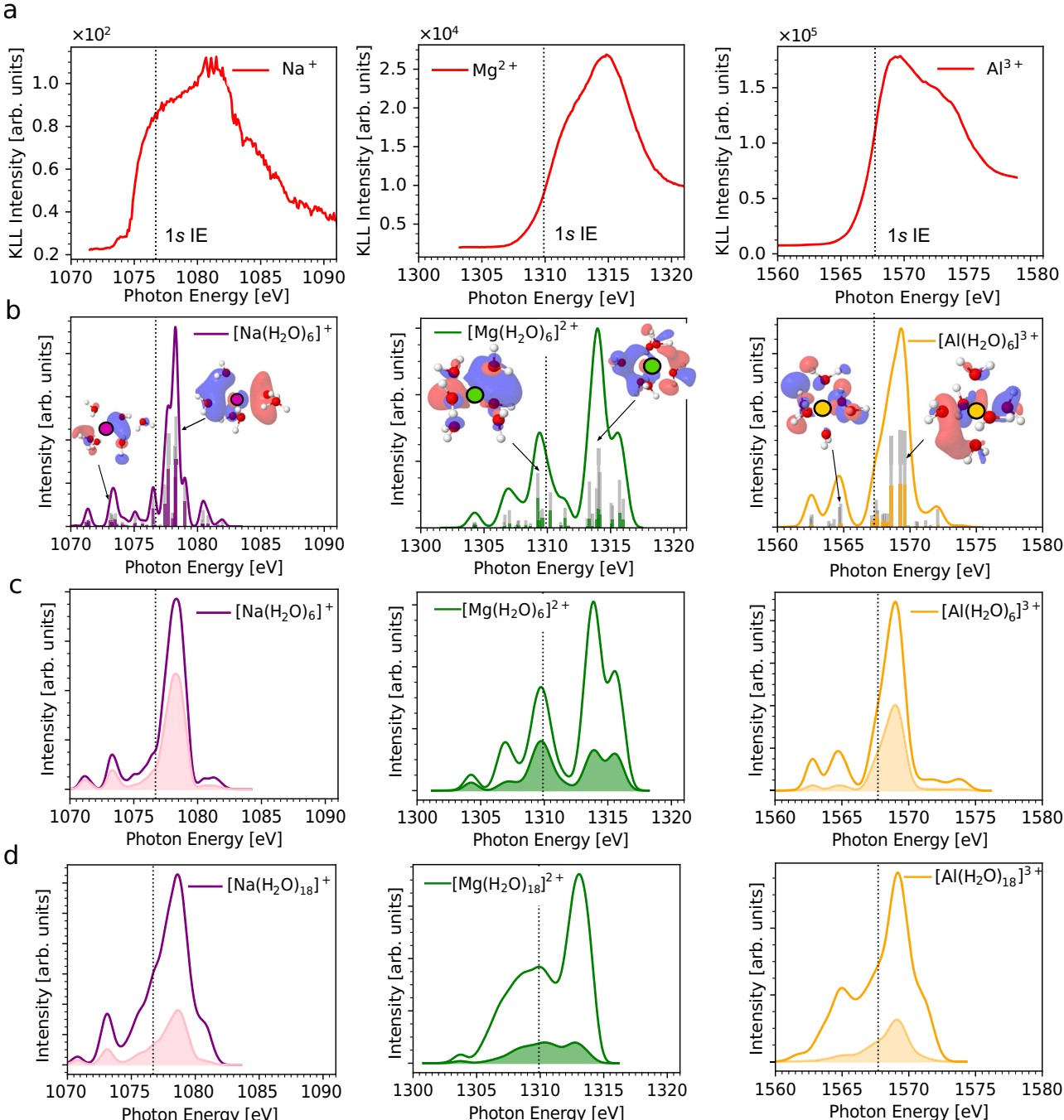

**Fig. 2 | Comparison between experimental and simulated XAS spectra.**
**a** Experimental KLL AM decay, partial-electron-yield XAS recorded in the vicinity of the aqueous Na+, Mg2+, and Al3+ cation 1s ionization thresholds (the spectra are based on the 2D plots, see Fig. 3). The vertical dashed lines indicate the 1s ionization energies (1s IEs). **b** Calculated XAS for a single structure of the [M(H2O)6]$^{n+}$ clusters at the SRC2-R2 level with a cc-pCVTZ basis set on the cations and a cc-pVTZ basis set on the water molecules in a polarizable continuum. Each calculated spectral point was broadened

by 0.28 eV for Na+, 0.43 eV for Mg2+, and 0.44 eV for Al3+. The gray bars show the oscillator strengths and the colored bars show the weights of the local contributions to the excited states. The most intense transitions are accompanied by corresponding NTOs as insets (purple−Na+, green−Mg2+, and orange−Al3+). **c** XAS spectra calculated for a set of 50 structures of [M(H2O)6]$^{n+}$ clusters. **d** XAS spectra calculated for a set of 50 structures of [M(H2O)18]$^{n+}$ clusters. The filled areas in panels (**c**) and (**d**) show the contribution of the local excited states to the overall spectrum.

means that the excited electron is closer to the metal center. The differences between individual cations in terms of $\sigma_e$ are much smaller for the larger cluster size [M(H2O)18]$^{n+}$ because the excited-state wave functions are also scattered over the second hydration shell.

The XAS spectra calculated for a set of 50 structures for the [M(H2O)6]$^{n+}$ clusters are shown in Fig. 2 panel (c) and for [M(H2O)18]$^{n+}$ clusters, which also include the second solvation shell, in Fig. 2 panel (d). The filled area shows the contributions of the local excited states

to the overall spectrum. As can be seen from the spectra, the local character is most pronounced for the Na+ cation (the filled area corresponds to 56% of the total area for [Na(H2O)6]+) and relatively weak for the Mg2+ and Al3+ cations for clusters including the first solvation shell (the filled areas correspond to 31% for [Mg(H2O)6]2+ and 36% for [Al(H2O)6]3+, respectively). When the second solvation shell is additionally included, Fig. 2 panel (d), the local contributions to the excited states are smaller for all three cations (24% for [Na(H2O)18]+, 14% for

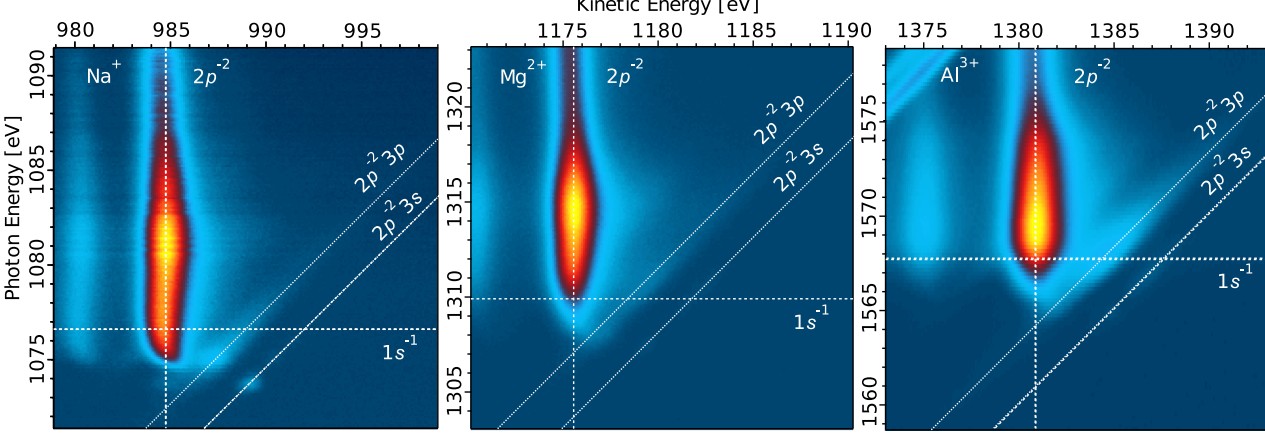

**Fig. 3 | 2D electron signal maps showing the electron kinetic energy as a function of photon energy, scanned across the cation 1s ionization thresholds, for Na$^+$, Mg$^{2+}$, and Al$^{3+}$ ions in water.** The vertical dashed and diagonal dotted lines indicate the $2p^{-2}$ final states (accessed after normal AM decay) and dispersive $2p^{-2}3p$ and $2p^{-2}3s$ final states (e.g., accessed after spectator AM decay), respectively. For Al$^{3+}$, the dispersive feature starting from about 1575 eV photon energy is due to the Cl$^-$ $2p^{-1}$ photoelectron peaks.

[Mg(H$_2$O)$_{18}$]$^{2+}$, and 15% for [Al(H$_2$O)$_{18}$]$^{3+}$. The highest local contributions are again observed for Na$^+$.

It is clear from the analysis of Fig. 2 that the cation charge and structural flexibility/rigidity of the environment shape the character of the CTTS states, which we use as a proxy of the electronic wave packet. The hydration environment likely similarly controls the actual outgoing electronic wave packet produced following excitation above the 1s ionization threshold.

**Identification of the decay channels and AM-type processes**

The next step is to study the decay channels following the X-ray absorption process. For that, we inspect the 2D photoemission data maps shown in Fig. 3 and follow the evolution of the resonant excitation and subsequent ionization of the 1s electron with increasing photon energy. In the 2D maps shown in Fig. 3, the $1s^{-1}$ ionization energies are marked by horizontal, dotted lines (the 1s ionization energies are detailed in the SI of ref. 36).

To interpret the possible decay channels, we employed ab initio calculations. The calculated energies corresponding to various electronic states are collected in Supplementry Table 1 and are shown in Supplementary Fig. 4 in the SI for Na$^+$, Mg$^{2+}$, and Al$^{3+}$ ions embedded in both water cluster sizes. Experimental energy values, as inferred from the 2D maps at the 1s ionization thresholds, are also presented in the Supplementry Table 1 in the SI. The best agreement with the experimental values is achieved when larger clusters are used for the MOM calculations; in the case of highly charged species in aqueous solutions, the long-range polarization of the environment is particularly important[36,38]. Based on the calculations, we assign the most intense feature in the 2D maps for all three ions to the KLL AM decay of the $1s^{-1}$ core-ionized states to produce $^1$D $2p^{-2}$ two-hole (2h) final states. This feature occurs at constant kinetic energy, corresponding to vertical spectral lines in each of the 2D maps shown in Fig. 3, specifically at 984.6, 1175.6, and 1380.9 eV kinetic energy for the hydrated Na$^+$, Mg$^{2+}$, and Al$^{3+}$ ions, respectively. All three 2D maps additionally show parallel but fainter constant-kinetic-energy signals at slightly lower kinetic energies (980, 1171, and 1376 eV for the hydrated Na$^+$, Mg$^{2+}$, and Al$^{3+}$ ions, respectively) corresponding to the transitions to another $2p^{-2}$ multiplet, $^1$S. In addition to the constant-kinetic-energy features associated with the KLL AM decay most importantly, a dispersive feature is also present for all three sample ions. A similar feature was observed in a previous study of aqueous-phase K$^+$ and Cl$^-$[37]. The kinetic energy of the dispersive feature is linearly dependent on the photon energy, meaning that the feature is characterized by a constant ionization

energy (88.0, 131.3, and 183.4 eV for the Na$^+$, Mg$^{2+}$, and Al$^{3+}$ ions, respectively). Based on the calculations, see below, we assign this dispersive feature to the $2p^{-2}3p$ final states after AM decay. These two-hole-one-electron (2h1e) states can be formed as shake-up satellite states to the main $2p^{-1}$ states (ionization energies of 35.4, 55.6, and 80.4 eV for the Na$^+$, Mg$^{2+}$, and Al$^{3+}$ ions, respectively[36]) and by spectator resonant AM decay from the 1s core-excited states, as depicted in Fig. 4. Additionally, as seen in Fig. 3, all three cations exhibit a parallel, low-intensity, constant-ionization-energy feature, which we assign to the $2p^{-2}3s$ (2h1e) final states after AM decay (spectator decay, also see Supplementary Table 1 in the SI).

As can be seen from Fig. 3, the KLL AM decay from the $1s^{-1}$ to the $2p^{-2}$ final states dominates both below and above the 1s ionization limit. The $2p^{-2}3p$ final states after spectator AM decay exhibit resonant enhancement even above the 1s ionization threshold. This can be understood in terms of the wave-packet evolution, see Fig. 4. After X-ray photon absorption, the electronic wave packet evolves into charge-separated states and, at the same time, decays via AM processes. Following near-resonant excitation, we can identify two associated main types of final states after AM decay: (i) the localized $2p^{-2}3p$ (2h1e) final states after spectator AM decay and (ii) the delocalized $2p^{-2}$ (2h) final states after electron delocalization and normal AM decay (see Fig. 4). Spectator AM decay to the $2p^{-2}3p$ final states requires that the excited electron has a sufficient degree of localized metal $np$ character. As the excited-state wave packet evolves from mostly localized metal $np$ character to the states delocalized over several water molecules, spectator AM decay to the $2p^{-2}3p$ final states mainly occurs for early-time AM decays. During the core-hole lifetime, the wave packet can evolve into states in which most of the electron density is on the surrounding water molecules, i.e., we observe electron delocalization from the cation to hydrating water molecules (see Fig. 4). Such later-time AM decays are not influenced by the excited electron, which is sufficiently delocalized and screened not to affect the local energetics of the metal ion, resulting in normal AM decay to the $2p^{-2}$ final states. These $2p^{-2}$ final states appear to be locally identical to the $2p^{-2}$ final states that are predominantly populated following non-resonant excitation well above the 1s ionization threshold, in which the ionized electronic wave packet is also delocalized from the metal ion and screened. The "electron delocalization" can be interpreted to represent the first step of charge separation and evolution from a localized excitation to a hydrated electron. Such competition between resonant AM decay and delocalization of the excited electron has also been observed in ref. 37.

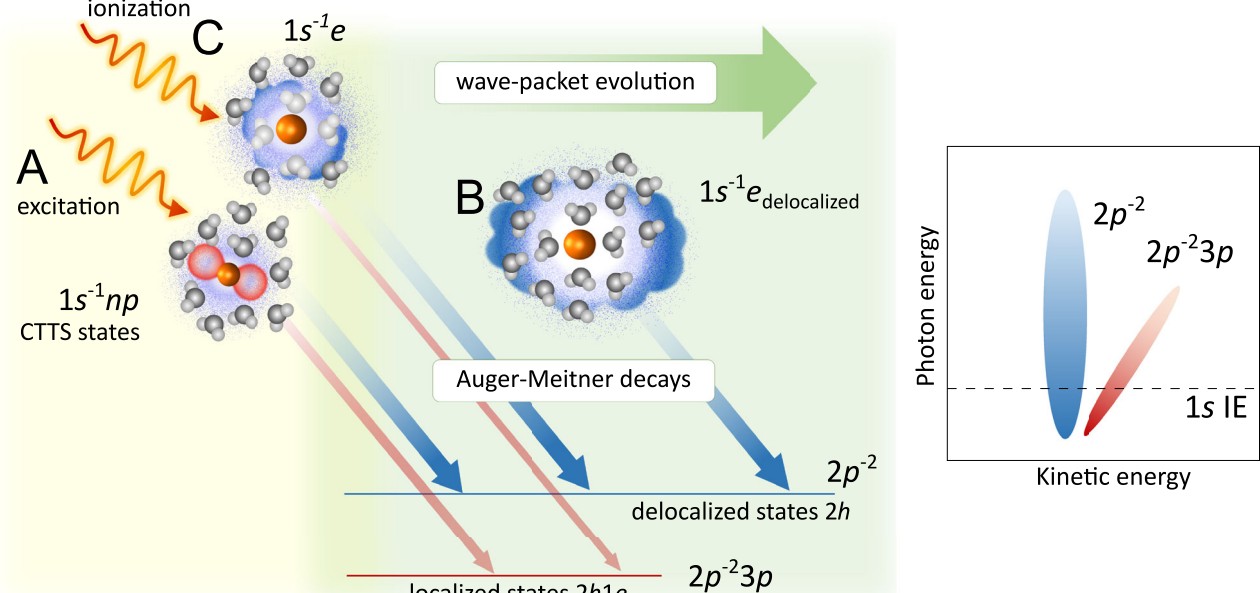

**Fig. 4 | Sketch of the discussed processes.** All processes start with an initial X-ray photoabsorption, as depicted on the left (light yellow area). The green arrow and central green section show the wave-packet evolution, which occurs simultaneously with the AM decay (light green area). Excitation of the $1s$ electron **A** yields a wave packet (denoted as $1s^{-1}np$ CTTS states) that is partially localized on the metal ion (red lobes) and partially delocalized on the surrounding water molecules (blue cloud). Associated early-time AM decays will reflect the more localized wave-packet character, resulting in either the localized final states after spectator AM decay, denoted as $2p^{-2}3p$ (red arrow), or the delocalized final states after normal AM decay, denoted as $2p^{-2}$ (blue arrow). Later AM decays take place with the wave packet being more delocalized on the surrounding water molecules (denoted as $1s^{-1}e_{\text{delocalized}}$, **B**), resulting predominantly in the $2p^{-2}$ final states (blue arrow). Above the $1s$ ionization threshold, photoabsorption of the $1s$ electron leads to an expanding photoelectron wave packet (denoted as $1s^{-1}e$, **C**), which again develops into $1s^{-1}e_{\text{delocalized}}$. In the early stages, AM decay partially leads to localized $2p^{-2}3p$ final states (red arrow, less likely) and predominantly to $2p^{-2}$ final states. In the later stages, the decay of $1s^{-1}e_{\text{delocalized}}$ into the $2p^{-2}$ final states dominates again. The right panel schematically shows the dependence of the delocalized $2p^{-2}$ (blue, vertical oval) and localized $2p^{-2}3p$ (red, diagonal oval) final states on the incoming radiation energy.

## Electron delocalization dynamics

The intensity ratio between the $2p^{-2}3p$ and the $2p^{-2}$ final states depends on the relative timescale of the electronic wave-packet evolution and the $1s$ core-hole lifetime. The $1s$ core-hole lifetime can be derived using $\tau_{1s} = \hbar/\Gamma_{1s}$, where $\Gamma_{1s}$ is the total decay width of that state. Based on ref. 39, experimental $\Gamma_{1s}$ parameters of $0.28 \pm 0.03$, $0.43 \pm 0.12$, and $0.44 \pm 0.04$ eV can be derived for metallic Na, Mg, and Al, respectively (where the Na value is associated with a single X-ray photoemission study and the Mg and Al values correspond to the averages and standard deviations of all of the tabulated experimental X-ray-spectroscopy data). Although these $\Gamma$ values are expected to vary somewhat with electronic charge state and chemical environment, they are generally found to be consistent with independent-particle-model computations[39] and our aqueous-phase Na$^+$, Mg$^{2+}$, and Al$^{3+}$ $1s$ data, as exemplified in Fig. S5. Based on the experimental values collected in ref. 39, the aqueous-phase metal $1s$ core-hole lifetimes can be inferred as $\tau_{1s}$ values of $2.35^{+0.28}_{-0.23}$ fs, $1.54^{+0.57}_{-0.33}$ fs, and $1.51^{+0.15}_{-0.13}$ fs for the Na, Mg, and Al $1s^{-1}$ states, respectively.

For a selected set of photon energies spanning the $1s$ resonant excitation to post-ionization regimes, the AM spectra were fitted, and the integrated intensities of the peaks corresponding to the $2p^{-2}$ and $2p^{-2}3p$ final states were determined. Note that there are two contributions to the $2p^{-2}3p$ final states: direct, off-resonant photoemission and resonant spectator photoemission. As we are not interested in the direct, off-resonant photoemission contribution, the intensity corresponding to this feature was subtracted from the integrated intensity. Using the intensities obtained in this way, the timescale of electron delocalization was determined as $\tau_{\text{delocalization}}$, which is calculated as $\tau_{1s} \times I_{2p^{-2}3p}/I_{2p^{-2}}$, where $I_{2p^{-2}3p}$ is the integrated intensity of the peak corresponding to $2p^{-2}3p$ and $I_{2p^{-2}}$ is that of $2p^{-2}$. Fig. 5 shows the correspondingly obtained $\tau_{\text{delocalization}}$ as a function of the photon energy

for the three ions, together with the associated standard deviations of the determined values (shaded areas), as determined by propagating the uncertainty in the reference $\Gamma_{1s}$ values for the different aqueous-phase cations.

As shown in Fig. 5, for Na$^+$, $\tau_{\text{delocalization}}$ is estimated to be ≈150 as at the $1s$ threshold, decreasing to ≈0 at 4.5 eV above the threshold; for Mg$^{2+}$, $\tau_{\text{delocalization}}$ is ≈200 as at the $1s$ threshold, decreasing to a value of about 20 as 4 eV above the threshold. In the case of the Al$^{3+}$ ion, $\tau_{\text{delocalization}}$ is longer than for Na$^+$ and Mg$^{2+}$, decreasing from about 350 as at the $1s$ threshold to about 100 as 3 eV above the threshold.

Comparing the results for the three ions, we see that $\tau_{\text{delocalization}}$ is similar for Na$^+$ and Mg$^{2+}$, and slightly longer for Al$^{3+}$ across almost the entire photon-energy range, except for at the lowest photon energies. There are likely both structural and energetic reasons for this observation. First, both the ion–water distance and the spatial extent of the excited states decrease from Na$^+$ over Mg$^{2+}$ to Al$^{3+}$[36]. This is supported by the picture provided by the ab initio calculations in Fig. 2 and Supplementary Fig. 1 in the SI.

Another important observation in the present study is that $\tau_{\text{delocalization}}$ decreases with increasing photon energy for all three ions. This is consistent with a higher degree of delocalization with increasing energy of the formed CTTS states. Yet, remarkably, the electron seems to reside in the vicinity of the cations even above the ionization limit. This points to the possible existence of long-living electron resonances in water under these conditions. Furthermore, it suggests that the core-excited state is a resonantly occupied state, weakly coupled to the surrounding bulk water via tunneling through an energy barrier[40,41]. In addition, we note that the electron delocalization into the bulk water requires empty states in water for the excited electron to transfer into. The XAS spectra in Fig. 2 represent the excitation of the $1s$ electron into unoccupied electronic states formed

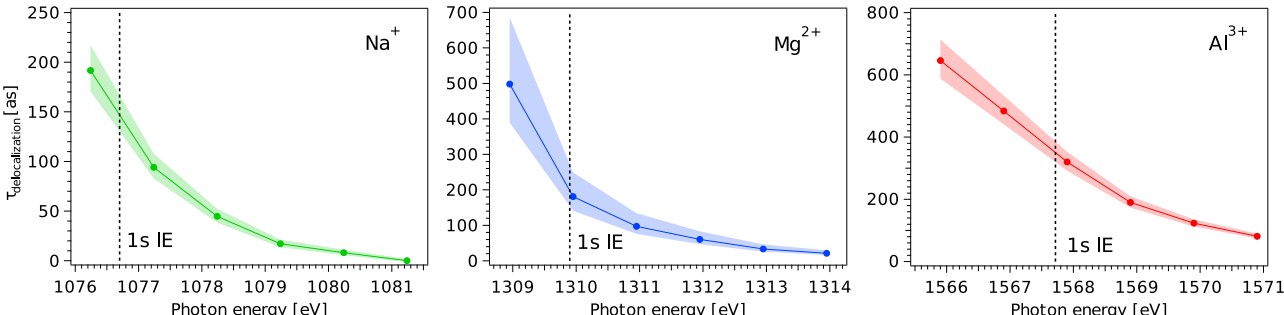

**Fig. 5 | The wave-packet-evolution timescales.** The timescales $\tau_{\text{delocalization}}$ were derived from the AM maps shown in Fig. 3 using the core-hole-clock method at selected photon energies. The shaded areas indicate the standard deviations of the determined values. The dashed lines indicate the 1s ionization thresholds of the respective aqueous ions, as determined from aqueous-phase 1s photoelectron peak fits[36]. We observe a continuous decrease of the wave-packet-evolution timescale from below to above the 1s ionization threshold.

by the unoccupied water valence levels in the liquid bulk (see refs. 42–46).

The calculated band structures[42–46] show a steep increase in the density of states with increasing energy, and the observed decrease of electron delocalization timescales with increasing photon energy is qualitatively consistent with an increased number of water states available for the excited electron to be transferred to. This enables rapid delocalization of the excited electron to unoccupied states in bulk liquid water, eventually leading to the formation of pre-hydrated and then hydrated electrons. Another perspective on these ultrafast processes is provided by the $Z+1$, or equivalent core, approximation. In this approach, the effect of the 1s core hole on the outer electrons is considered to be like that of an extra proton in the nucleus. This means that the valence electronic structure of an element $Z$ with a core hole will be like that of an element $Z+1$. In this case, this means that the core-excited states of $Na^+$, $Mg^{2+}$, and $Al^{3+}$ ions will temporarily resemble the ground states of $Mg^+$, $Al^{2+}$, and $Si^{3+}$ ions, respectively. While Si does not form monoatomic ions in water, Mg and Al do. However, $Mg^+$ and $Al^{2+}$ are not stable ions in water and would rapidly give up electrons to form $Mg^{2+}$ and $Al^{3+}$. For these two cations, the $Z+1$ approximation provides a qualitative understanding of our observation of electron delocalization dynamics after core excitation.

## Conclusions

Ultrafast electron dynamics connected to core-level excitation and associated formation of the CTTS states of the isoelectronic $Na^+$, $Mg^{2+}$, and $Al^{3+}$ ions in water have been investigated using a combination of X-ray absorption spectroscopy, AM spectroscopy, and ab initio calculations. Due to the short core-hole lifetime of about 2 fs, the AM spectra reflect the ultrafast dynamics of the system. The 1s excitation of the metal ion produces electronic wave packets partially localized on the ionized atom, which gradually evolve towards states with electron density delocalized over the surrounding water molecules. This "electron delocalization" represents the first step of charge separation in the evolution from a localized excitation to a hydrated electron.

In the AM spectra, final states after AM decay of $2p^{-2}3p$ and $2p^{-2}$ character are identified, and both types are observed both below and above the 1s ionization threshold. As no qualitatively different behavior is observed below and above the 1s threshold, it is inferred that the slow outgoing photoelectron wave resides in the vicinity of the core-ionized ion for a sufficient amount of time that the $2p^{-2}3p$ final states can be formed. The relative intensity of the $2p^{-2}3p$ and $2p^{-2}$ final states reflects the wave-packet evolution from more localized $np$ states to delocalized states, with most of the electron density localized on the water molecules. Using the core-hole-clock method, we see that the delocalization timescales decrease from $\approx 150$ as for $Na^+$, $\approx 200$ as for $Mg^{2+}$, and $\approx 350$ as for $Al^{3+}$ below the 1s ionization threshold to $\approx 20$ as for $Na^+$, $\approx 20$ as for $Mg^{2+}$, and $\approx 100$ as for $Al^{3+}$ above the threshold.

It is remarkable that no discontinuity in the decrease of $\tau_{\text{delocalization}}$ is observed at the 1s ionization energies. The $2p^{-2}3p$ (2h1e) states are, in fact, also enhanced in intensity for the photon energies exceeding the 1s ionization threshold. At these photon energies, the electron is no longer bound, but it is quite slow and possibly temporarily trapped by scattering in the local potential, as observed for molecular shape resonances[47]. We speculate that this may lead to electron recapture after the AM decay, explaining the enhanced intensity just above the 1s threshold.

The picture provided by the ab initio simulations is not complete, yet it reveals an important insight into the differences between the hydration patterns and electron-delocalization-to-solvent feasibility of the respective ions. Full ab initio dynamics, taking into account the gradual formation of the initial wave packets and simultaneous AM decay on the femtosecond timescale in a condensed environment, is beyond state-of-the-art computational capabilities. Consequently, the present systems can serve as convenient test beds for the further development of quantum dynamics calculations of complex hydrated systems. Recently, theoretical progress has been obtained for related phenomena connecting the simultaneous appearance of photoionized and autoionized electrons, such as Fano resonances[19,48,49].

The present results on X-ray-induced electron delocalization provide insight into the ultrafast dynamics of the first step of charge separation in aqueous solutions and into the process of going from a localized excitation to a hydrated electron. The electron delocalization can be seen as an example of an intermolecular electron-transfer process, which is a fundamental step in many chemical processes. With the advent of attosecond X-ray pulse sources, such as free-electron lasers (FELs), it is becoming possible to study the dynamics of these and other types of X-ray-induced processes through direct X-ray pump−X-ray probe measurements. Our results thus provide a glimpse of one of the complex phenomena that ultrafast science is now starting to probe, although the application of attosecond photoemission pump-probe schemes to the aqueous phase using FELs remains technically challenging.

## Methods
### Experimental setup

The AM electron spectra for the hydrated $Na^+$, $Mg^{2+}$, and $Al^{3+}$ ions were measured across the metal-atom 1s ionization energies (IEs) at the P04 beamline of the synchrotron facility PETRA III (DESY, Hamburg, Germany),[50] using the *EASI* liquid-microjet photoemission spectroscopy setup[51]. A similar $Mg^{2+}$ spectrum was recorded at the HIPPIE beamline of MAX IV (Lund, Sweden)[52] (see Supplementary Fig. 5 in the SI).

The present analysis is based on the spectra recorded at P04. The P04 beamline provides circularly polarized X-rays with energies between 250−3000 eV and an on-target flux of $\approx 10^{12}$ photons/s (at a

resolving power of 10,000). Here, the beamline was operated with a 1200 lines/mm grating and a 100 μm exit-slit width, offering photon-energy resolutions between 210 and 380 meV at photon energies between 1050 and 1600 eV. A general description of the *EASI* experimental setup is given in ref. [51]. With an 800 μm entrance slit into our hemispherical electron analyzer and 100 eV ($Na^+$ and $Mg^{2+}$ samples) or 200 eV ($Al^{3+}$ sample) pass energies implemented in these measurements, analyzer energy resolutions of ≈200 and ≈400 meV were respectively achieved. This led to estimated total experimental energy resolutions of ≈290, ≈360, and ≈560 meV in the $Na^+$, $Mg^{2+}$, and $Al^{3+}$ experiments, respectively. The sample solutions were horizontally injected into the sample vacuum chamber using an HPLC pump (0.8 ml/min flow rate) and a fused silica nozzle (inner diameter of 28 μm). This resulted in a free-flowing, cylindrical liquid microjet and a ≈$10^{-4}$ mbar average pressure in the interaction chamber. The generated liquid microjet was irradiated by the focused synchrotron radiation (≈180 × 40 μm², horizontal × vertical, spot size) 1–2 mm downstream from the tip of the jet nozzle. The kinetic energies of the emitted electrons were measured using a differentially pumped hemispherical electron analyzer, mounted at a 130° angle with respect to the X-ray beam (back-scattering geometry[51]), which is close to the magic angle. The distance between the ionization locus and the 800-μm entrance orifice of the electron analyzer was less than 0.5 mm, which results in a ≈±20° acceptance angle, with the exact value depending on analyzer operation mode[51]. The aqueous solutions of 2 M $AlCl_3$, 1 M $MgCl_2$, and 1 M NaCl were prepared by respectively dissolving commercially purchased $AlCl_3$, $MgCl_2$, and NaCl (Sigma–Aldrich with purity >98%) salts in MilliQ water (18.2 MΩ/cm).

For each sample, the electron spectra were measured as a function of the photon energy across the $1s$ ionization thresholds of the cations. These series of photoemission spectra are presented as two-dimensional (2D) maps of photon energy versus electron kinetic energy, with the intensity of the emitted electrons encoded in a color scale. This is a common presentation (see Fig. 3 of the present work), which provides a convenient visualization of evolving resonant spectral features[53,54]. Single photoemission (i.e., AM) spectra, recorded at a given photon energy, then correspond to a horizontal cut through a given map of the type shown in Fig. 3. Signal integration in the horizontal direction, within the kinetic-energy range of the AM peaks, yields the respective partial-electron-yield X-ray absorption spectra as a function of photon energy, PEY-XAS. The individual electron spectra were fitted using the Spectrum Analysis by Curve Fitting (SPANCF)[55] macro package for Igor Pro (Wavemetrics, Inc., Lake Oswego, USA). The main KLL AM peaks have a high-kinetic-energy tail arising from post-collision interaction (PCI) between the photoelectron and the AM electron. Thus, PCI profiles were used to fit the AM peaks, while Voigt profiles were used to fit the $2p^{-2}3p$ peaks. The integrated intensities and peak positions of the aforementioned peaks were obtained from these fits. An exemplary fit for the case of the $Al^{3+}_{(aq)}$ spectrum recorded at a photon energy of 1569.9 eV is shown in Supplementary Fig. 7, and the obtained energies and intensities for all analyzed cases are listed in Supplementary Table 2. The photon energies and the electron kinetic energies were calibrated using peaks of known ionization energy in combination with the first- and second-order light from the beamline, as described in detail in ref. [36]. Photoemission spectra recorded at fixed photon energies above the respective $1s$ thresholds were analyzed in our previous works refs. [36,56] to reveal signatures of non-local solute autoionization channels, which occur at higher or lower kinetic energies than the data reported in this work.

## Theoretical methods
The simultaneous ultrafast processes interrogated here were divided into individual steps. The initial configurational space was sampled by classical molecular dynamics (MD) simulations. The simulation box for NaCl and $MgCl_2$ solutions contained 160 molecules of salts and an appropriate number of water molecules (8753 water molecules for NaCl and 8810 for $MgCl_2$). For $AlCl_3$, the simulation box contained 267 aluminum cations, 801 chloride anions, and 8472 water molecules. The total length of each simulation was 200 ns, the time step for the propagation was set to 2 fs, and 3D periodic boundary conditions were employed. The simulation temperature was set to 300 K and was controlled by a velocity-rescale thermostat with time coupling set to 0.5 ps (0.1 ps for $AlCl_3$). The pressure of the system was set to 1 bar, which was controlled by the Parrinello–Rahman barostat with a coupling constant of 1 ps (2 ps for $AlCl_3$). The van der Waals interactions were truncated at 1.5 nm (1.2 nm for $AlCl_3$); the long-range electrostatic interactions were calculated by the particle mesh Ewald method. The details of the force field are described in great detail in ref. [36]. The simulations were performed for 2 M $AlCl_3$, 1 M $MgCl_2$, and 1 M NaCl solutions to match the experimental conditions. For further calculations, we selected 200 or 50 geometries for two cluster sizes: $[M(H_2O)_6]^{n+}$ and larger $[M(H_2O)_{18}]^{n+}$, where $n = 1, 2, 3$ for M = Na, Mg, Al, respectively. The smaller cluster size represents the first hydration shell around the solvated cation, and the larger cluster includes the hydration shell up to the second layer.

## X-ray absorption
The X-ray absorption spectra (XAS) were calculated at the equation-of-motion-coupled-cluster level with a core-valence separation scheme (CVS-EOM-EE-CCSD) and the time-dependent density functional theory (TDDFT) level with a restricted orbital space. Both methods can only provide stationary states of the studied systems, ignoring the coupling to the continuum, and can only provide proxies of the true XAS spectra. The spectra at the CVS-EOM-EE-CCSD level were constructed for an ensemble of 50 geometries of $[M(H_2O)_6]^{n+}$ in the gas phase. The calculations of the core-excited states require relatively large basis sets, preferentially with triple-$\zeta$ quality. In the case of CVS-EOM-EE-CCSD, a combined basis set—the uncontracted cc-pCVTZ on the metal cation and the smaller cc-pVDZ on the water molecules—was used. The TDDFT calculations were performed for both cluster sizes, including the first and also the second solvation shells, for an ensemble of 50 structures of $[M(H_2O)_6]^{n+}$ and $[M(H_2O)_{18}]^{n+}$, each. The SRC2-R2 functional, specifically tailored to core excitations[57], with the cc-pVTZ basis set on water and the cc-pCVTZ basis set on the metal cation, was employed throughout the simulations. The spectra were broadened by 0.28 eV for $Na^+$, 0.43 eV for $Mg^{2+}$, and 0.44 eV for $Al^{3+}$ to reflect the core-hole lifetime broadening[39]. XAS spectra were also simulated within the real-time time-dependent functional theory approach (RT-TDDFT) with the SRC2-R2 functional[58] and the combined cc-pVTZ and cc-pCVTZ basis sets[59]. The non-equilibrium solvation model, as implemented in Q-Chem 6.0[60], was used for all TDDFT XAS calculations.

Some properties of the excited states in the complex environment can be better understood in terms of a two-body exciton wave function, which describes the correlated motion of the hole and electron quasiparticles[61,62]. Specifically, we focused on decomposing the excited states into local contributions (localized on the metal cation) and charge-transfer contributions (localized on the water molecules). For this purpose, the so-called charge transfer variable, CT, was calculated. This collective variable quantifies the total weight of configurations for which the initial and final orbitals are situated on different fragments, e.g., localized on the surrounding water molecules. The definition of the variables and a detailed discussion can be found in Supplementary Note 1 in the SI. The analysis of the excited-state wave functions was performed by means of natural transition orbitals (NTOs) and exciton analysis[63] using the TheoDORE code[64] and the wave function analysis library (libwfa)[61,62], to directly analyze the one-particle transition density matrix and to compute the exciton and electron sizes and the charge transfer numbers. Charge transfer numbers are based on population analysis and describe the weight of the transitions in which

the electron density is transferred from one fragment to another, in the current case, from the metal ion to water molecule(s).

## Energetics of the final states after AM decay

In the solvent, the final states after AM decay can be conveniently described by the maximum-overlap method (MOM)[65]. The tailored short-range functionals exhibit larger errors for the MOM calculations[58,66]. Therefore, the $\omega$B97X-D functional was selected with the cc-pVTZ basis set on all atoms. The implementation of the MOM method provides only triplet-state energies for doubly ionized states; the energy differences between states of various multiplicities were evaluated at the multireference configuration interaction (MRCI) level with an aug-cc-pVTZ basis set.

## Solvent effects

The non-equilibrium solvation model was used for the description of the final states after AM decay via the MOM. The permittivity of the solution was adjusted for 1 and 2 M concentrations[36]. All calculations were performed in Q-Chem 6.0[60].

## Data availability

The data generated in this study have been deposited in the Zenodo database under accession code DOI: 10.5281/zenodo.10600582. All data needed to evaluate the conclusions in the paper are present in the paper and/or in the Supplementary Information.

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

## Acknowledgements

We acknowledge DESY (Hamburg, Germany), a member of the Helmholtz Association HGF, for the provision of experimental facilities. Parts of this research were carried out at PETRA III and we would like to thank Moritz Hoesch and his team for assistance in using beamline P04. Beamtime was allocated for proposal I-20190339. We acknowledge Jens Buck, Florian Diekmann, Matthias Kalläne, and Sebastian Rohlf from Christian-Albrechts-Universität zu Kiel for providing us with software to scan the photon energy via our SCIENTA spectrometer software. We thank Sebastian Malerz for participating in some of the measurements. We acknowledge Andrey Shavorskiy and his team, as well as the MAX IV Laboratory, for time on the beamline HIPPIE under proposal 20180386. Research conducted at MAX IV, a Swedish national user facility, is supported by the Swedish Research Council under contract 2018-07152, the Swedish Governmental Agency for Innovation Systems under contract 2018-04969, and Formas under contract 2019-02496. B.W. and U.H. acknowledge funding from the European Research Council (ERC) under the European Union's Horizon 2020 research and innovation program under Grant Agreement No. 883759-AQUACHIRAL. P.S. and E.M. thank the Czech Science Foundation for the support via project number 23-07066S. F.T. acknowledges funding by the Deutsche Forschungsgemeinschaft (DFG, German Research Foundation)—Project 509471550, Emmy Noether Program. F.T. and B.W. acknowledge support by the MaxWater initiative of the Max-Planck-Gesellschaft. C.C. acknowledges the Swedish Research Council (project 2018-00740) and the Helmholtz Association through the Center of Free-Electron Laser Science at DESY. O.B. acknowledges funding from the Swedish Research Council (VR) for the project VR 2023-04346.

## Author contributions

G.G., I.U., G.Ö., and O.B. conceived the experiments. G.G., I.U., G.Ö., D.C., F.T., I.W., E.C., U.H., B.W., C.C., and O.B. planned, prepared, carried out the experiments, and discussed the data. E.M. performed the theoretical calculations. E.M. and P.S. interpreted theoretical data. G.G., U.H., and G.Ö. analyzed the data. G.G., E.M., and O.B. wrote the manuscript with feedback from all authors.

## Funding

## Competing interests

The authors declare no competing interests.
