## [Peer Review File · Nature Communications]

Attosecond formation of charge-transfer-to-solvent states of aqueous ions probed using the core-hole-clock techniqueREVIEWER COMMENTS

Reviewer #1 (Remarks to the Author):

The manuscript from Muchová et al presents an investigation on the dynamics of solvated electrons, generated by X-ray absorption of ions with different ion charges in aqueous solutions.

The experimental results based on Auger-Meitner photoelectron spectroscopy interpreted within the core-hole-clock technique, allow the estimation of the so-called electron "delocalization time". This process is the first step towards the formation of solvated electrons and charge-transfer-to-solvents (CTTS) states.

The experimental results are supported by three different model calculations, presented in Fig. 1b of the main text and Fig. S2a and S2b. These calculations are done within different approximations and each of them for two different cluster sizes, which include only the first and up to the second solvation shell, respectively.

The main experimental result presented in the manuscript is the estimation of the delocalization times for three different salts in solution. The timescales, ranging between a few tens to a few hundreds of femtoseconds for different photon energies across the 1s ionization threshold, are similar for the three samples. Not unexpectedly longer timescales are observed for higher charged ions (Al^{3+}) and the authors comment and discuss this finding on the basis of structural and energetic reasons.

Another important result, common to all the samples investigated, is the decrease of the electron delocalization time with increasing energy.

The research topic presented in the manuscript is timely and significant for the broad readership of Nature Communications, as demonstrated by a very recent paper, which was published in Nature Comm, maybe even after the current manuscript was submitted to the same journal (Lan et al, Nat Comm 15, 2544 [2024]).

This latest paper together with the references listed by the authors in the manuscript show that the investigation of electron solvation dynamics and CTTS formation in general has become a research topic of great relevance within the physical chemistry science community.

In this context, the present manuscript adds the significant contribution of investigating the very first step involved in the metal-to-solvent electron transfer process. In fact, while the formation of CTTS and their consequent relaxation that brings to the solvated electrons final state has been studied and characterized extensively, little has been reported about the initial excitation and delocalization process.

For all these reasons, I believe that this work deserves publication in Nature Communications. However, before giving my full recommendation, I would like the authors to comment on the following points:

- My main concern is about the simple formula reported at the end of page 15 used to extract the wavepacket evolution time constant t_{WPE} from the electron spectra.

I'm not sure whether I miss something, but I believe that the formula is not consistent with the observations and the plots reported in Fig. 5.

In fact, as it is currently written, t_{WPE} increases the larger is the ratio between the intensities of the $2p^{-2}$ (from delocalized states) and $2p^{-2}3p$ peak (from localized states).

From Fig. 3, it is apparent that for all the photon energies, the intensity of $2p^{-2}$ peak (not dispersive) is always larger than the $2p^{-2}3p$ peak (dispersive), and this is true for all the samples investigated. Consequently, the formula would return a value of t_{WPE} larger than t_{1S} basically for all photon energies, since the ratio is evidently always larger than one.

Additionally, also the photon energy trend shown in Fig. 5 seems not consistent with the formula for t_{WPE} .

Let's have a look more in details at Fig. 3 for the Na^+ case: at 1075 eV (below 1s threshold), from the 2D plot the $2p^{-2}$ peak is "relatively weak". There is not a color scale, but the the maximum

intensity of the peak is for photon energies above the 1s threshold, as it should be. On the other hand, the $2p^{-2}3p$ peak is faint, but its intensity can be considered comparable, at this photon energy, to the one of the $2p^{-2}$ peak.

On the contrary, at 1080 eV (above the 1s threshold), the intensity of the $2p^{-2}$ peak is "large" and the one of $2p^{-2}3p$ peak is very small, making the ratio in the formula a very big number. As such, one would calculate that $tWPE_{1075eV} \ll tWPE_{1080eV}$ but Fig 5 shows the opposite. The trend obtained from the formula would be also contrary to the intuitive picture that higher energy induces a shorter electron delocalization time.

I would like the authors comment on these observations. Maybe to make things clearer, the authors could show linecuts of the 2D plots at two specific photon energies (for instance 1075 and 1080 eV for Na⁺ mentioned above) so that the readers can have an idea about the respective integrated peak intensities?

- Regarding the introduction, can the authors elaborate more in details why "indirect" techniques like core-hole-clock methods are advantageous with respect to "direct" ultrafast spectroscopy performed with attosecond pulses when it comes to investigate liquid samples? In the introduction, it is stated that the requirements are high rep-rate, low pulse energy, attosecond pulses. My understanding is that these are exactly the parameters matched by High order Harmonic Generation (HHG) sources. Admittedly the latter are not easily providing up to the soft X-ray photons, but in the literature there are a few examples of sources able to do that. Otherwise, soft X-rays FELs represent also a valid alternative.

- If possible, it would be nice to have an exemplary figure representing the extraction of the information as presented in the text at the end of page 6 / beginning of page 7. Something like a line cut of the 2D plot with the PCI and Voigt curves used to fit the delocalized ($2p^{-2}3p$) and localized ($2p^{-2}$) states peaks.

- I appreciated the presentation of Fig.2, showing the molecular orbitals and identifying where are the unoccupied states that would most likely host the delocalized electrons. But it is said that those orbitals have been calculated for the most intense transition below the 1s ionization threshold.

Would this transition be in correspondence with one of the peaks seen in Fig 1b? If this is the case, these peaks are somehow visible for the small cluster (with 6 H₂O molecules, green solid line) but not so much for the large one (with 18 H₂O molecules, black solid line), for which the calculations show a broad spectrum without narrow peak features, in particular for Mg²⁺ and Al³⁺ ions. How is the peak identified then?

Also for Al³⁺ case, apparently, the most intense peak for the small cluster is at a photon energy a few eV lower than the most intense "peak" of large cluster. Is possibly this the reason for the quite different electron density maps shown in Fig. 2c?

- For completeness, since it is mentioned in the text and reported in Table S1, I suggest adding the $2p^{-2}3s$ dispersive line to Fig. 3 as well.

Furthermore, these states seem to show resonant enhancement above the 1s ionization threshold even beyond the $2p^{-2}3p$ states. Can the same analysis on the $2p^{-2}3p$ be performed also on the $2p^{-2}3s$ states? Would the conclusions be the same? Probably the qualitative trend (delocalization time getting smaller for higher energies) yes, but quantitatively also?

- A last final comment about data treatment and methodology: how was the off-resonant photoemission contribution subtracted by the integrated intensity of the localized states?

Minor remarks:

- Typo in Table S1: Na⁺ is indicated with double charge (2+) as Mg.

- Fig S5: I think the caption does not reflect what is shown in the figure, there are no plots of the residuals from the fit, neither the green baseline is visible in any of the panels (only the red data and the blue fits are shown).

Reviewer #2 (Remarks to the Author):

E. Muchova et. Al. studied the electron transfer mechanism that is important to chemical reactions in aqueous solutions. This important process involves a core-level excitation of the 1s electron from metallic ion to the intermediated electronic states followed by the electron transfer to the delocalized neighboring solvent water molecules. The lifetime of the charge transfer of valence electron is estimated using so-called core-hole clock measurements associated with the absorption width and kinetic energy of Auger-Meitner (AM) electrons. The group has extensive expertise and knowledge on this field and the manuscript is submitted to Nature Communication for a broader readership. From my point of view, this is a good study in general. There are several aspects that require clarification before publishing in Nat. Comm. General remarks: Authors should carefully address which electron(s) is described since the dynamics involved in core and valence electrons at the same time. In this manuscript, the dynamics terms, for example, electronic wavepacket and charge transfer are used for valence electrons, and the AM process and core-hole clock are for core electrons. These are used to describe the processes of dynamics. The "state" is usually used to describe the electronic configuration as a whole (including metal ion and water). Throughout the whole manuscript, I can tell that Authors mixing these all together and this may be a very confusing explanation for a broader readership. All the detailed comments can be found in the attached PDF document.

1. Minor suggestions are shown in the attached PDF document.
2. Although you provided the lifetime of the core-excited metallic ion in water, the most attractive part of your study is that you are able to determine the core-excited CTTS of metallic ion in aqueous solution. From your title, it is hard to link any attracting feature that directly related to your study. You may need to consider a better title.
3. All the orbitals should be in italic form, for example, 1s should be *1s*.
4. Actually, I believe the importance to Authors' studies is in the field of the Radiolysis of liquid water (similar to ref 34 that brings up the related field of radiochemistry) and associated charge transfer dynamics, especially with ions in water irradiated by the high energy X-ray. This study here actually provide very valuable information of the early stage of radiolysis of aqueous solution containing different metal ions, which may eventually lead to the corrosion of the nuclear power plant and container of nuclear waste etc. If abstract and introduction can have some touch on this aspect, it will raise more attention and impact to this publication. This may eventually require you to change your title accordingly.
5. In INTRODUCTION and reference 2-11, these are charge-transfer from the valence excitations, and it is different from your study which involves in the core-excitation process (followed by CTTS). It will be clear for readers to understand your study with a diagram to point out how you excite target system and which electron is transfer and how this is different from the charge transfer from valence excitation.
6. Please also cite this paper: Science 383, 1118–1122 (2024); this is the first Attosecond X-ray pump and attosecond X-ray probe on liquid water. (using XFEL). It will be after your Ref 16.
7. In "EXPERIMENTAL SETUP" section, Authors explain the 2D, however, there is no figure to help description of the details. It will be helpful to readers that Figure 3 is moved to the front of the manuscript and incorporated with the current Figure 1 (since it is a partial or full projection of the current Figure 3). The detailed data fitting associated with AM peaks (PCI, resonant and non-resonant) should be presented in SI. These are missing.
8. "THEORETICAL METHODS" should be moved to SI and only a few sentences are needed to summarize the experimental support by calculations.
9. In "RESULTS AND DISCUSSIONS" section, the description in "X-ray absorption. We start with ..., thereby reflecting the electron dynamics" seems to work better in INTRODUCTION for readers. This part is highlighted yellow in the attached PDF.
10. I have a trouble understanding the "final decay states" though out the manuscript. Do you mean it is the final state of AM process? Or even this state will decay to different state(s)? Please clarify this.
11. In the caption of Figure 1, there is no explanation of 0.2 eV phenomenological value. Why 0.2 eV? And justification?
12. The theory spectral shown in Figure 1(b) seems to have no help to Figure 1(a). I cannot see any resemblance between the data and calculations. And why only 6 and 18 waters are used to calculate spectra? This needs to be clarified.

13. All the "panel (b) of Figure 1" should be changed to "Figure 1(b)" if possible.
14. The legend boxes in Figure 1 overlap with figure frames in 1(a) Mg²⁺ and 1(b)Al³⁺. In addition, the X axis only has 3 minor ticks for 5 eV span. This is not easy to locate energy positions.
15. Figure 2: need to specify your metal color in the caption.
16. Please clarify what the electron size is (physical picture)? I know you have that in equation shown in Theory part, but it is not clear in terms of physical picture of this parameter.
17. Please mark 2p-23s line in the current Figure 3.
18. The axes and ticks for Figure 3 are very hard to read. Please change it to a thicker axes and ticks, and maybe consider different color that is different from the dark 2D map.
19. The "contributions" shown on the 4th line in "Electron delocalization dynamics" is not clearly specified. In addition, I can not understand why at this moment Authors start to discuss how core-hole clock works. This should be moved to introduction section.
20. The parameter " Γ " is not defined at the beginning in the manuscript. M is not defined either.
21. The intensity ratio between 2p-23p and 2p-2 peaks should be presented in the manuscript (with peaks shown Figures) and details analysis should be in SI. These are lacking.
22. The time constant τ_{WPE} is not defined at the beginning of its appearance. Does WPE represent "wavepacket evolution"? I have to guess it.
23. Authors used ionization thresholds for IE and BE throughout the manuscript (figures). Please be consistent.
24. What kind of tunneling do you mean on page 17? The electron tunneling during the CTTS? I cannot understand how your Figure 2 supports Figure 5 on page 17. There is no explanation of the colored shaded area following the data lines. Error bars? If so, what confidence levels?
25. It is striking to me that it has longer CTTS time scales when water molecules are closer to metal ions. Can you clarify it? Is that because it contains more metallic character when waters bond to metal ions and those bonding does not speed up the electron transfer to the outer free water molecules? This means that the ratio between 2p-23p and 2p-2 peaks will be larger for Al³⁺ case?
26. A figure will be needed to explain why the speed up transfer of electron for all ions in water shown in Figure 5 and on page 18. How does the empty water states overlap with your excited water wavepacket (in a figure). This is an important information from your studies.
27. This statement is intuitive and good to let reader know it. I was wondering if you are able to have calculation to support the observed time constant ore trends? So far, there is no theory support on your observation time scale as a function of photon energy, right?
28. Can you also provide an outlook on: how the attosecond pump and attosecond probe at FEL help to verify your measurements?

Reviewer #3 (Remarks to the Author):

The authors report an interesting study coupling attosecond pump probe measurements with static quantum chemical calculations to evaluate the charge transfer time of mono-, di- and tri- charged ions in aqueous solution after inner layer ionization (1s shell). They measure the rate of wave packet delocalization in less than a femtosecond. This work sheds light on the stages preceding electron solvation.

I find the writing of the manuscript not very didactic and complicated to follow for a non-specialist. It seems to me that proofreading could help improve this point. I'm not an expert on the experimental part, which I'm not in a position to criticize outright. However, my reservations about the current version of the manuscript (see below) lead me to recommend resubmission with major modifications.

The discussion of Figure 2 on page 12 is not very clear from my point of view. I find it hard to be convinced by the representation of the 3 NTOs in the Figure of the presence of a cage around Al³⁺. Moreover, this analysis is based on a single conformation, right? As the authors did calculations on 200 geometries, it would be useful to capture this diversity in the plot?

I'm surprised that the photoelectron stays in the vicinity of the cation long enough. I would expect it to escape further away ? A dynamical description is somehow missing to interpret fully the data. I agree that simulating the process delocalisation in real time is a challenge for theory, but this is where simulations would make sense. Shouldn't this challenge be taken up for an article in Nat. Commun? Considering fixed core positions (static calculations) may be a problem, even on such short time scales. The remarkable H₂O⁺ simulations published by several groups in recent years (notably the Czech school) have shown a close coupling between core dynamics and the degree of gap delocalization. This makes me wonder about the present case.

The theoretical part lacks a great deal of technical detail to enable us to pinpoint the type of simulation carried out and the robustness of the results obtained. I'm not claiming that the calculations were poorly done, but referring to reference 34 without explanation isn't enough. The same applies to the absorption X. These details could be placed in SI. They should enable a third-party researcher to reproduce the article's results.

Furthermore, which simulation results are taken from reference 34 and which are new to the present article? This is not made explicit in the manuscript.

Page 3 : EUV is not defined

Revision made to manuscript "The conception of a hydrated electron: X-ray-induced attosecond electron dynamics of aqueous ions"

Reviewer 1

The manuscript from Muchová et al presents an investigation on the dynamics of solvated electrons, generated by X-ray absorption of ions with different ion charges in aqueous solutions.

The experimental results based on Auger-Meitner photoelectron spectroscopy interpreted within the core-hole-clock technique, allow the estimation of the so-called electron "delocalization time". This process is the first step towards the formation of solvated electrons and charge-transfer-to-solvents (CTTS) states.

The experimental results are supported by three different model calculations, presented in Fig. 1b of the main text and Fig. S2a and S2b. These calculations are done within different approximations and each of them for two different cluster sizes, which include only the first and up to the second solvation shell, respectively.

The main experimental result presented in the manuscript is the estimation of the delocalization times for three different salts in solution. The timescales, ranging between a few tens to a few hundreds of femtoseconds for different photon energies across the 1s ionization threshold, are similar for the three samples. Not unexpectedly longer timescales are observed for higher charged ions (Al³⁺) and the authors comment and discuss this finding on the basis of structural and energetic reasons.

Another important result, common to all the samples investigated, is the decrease of the electron delocalization time with increasing energy.

The research topic presented in the manuscript is timely and significant for the broad readership of Nature Communications, as demonstrated by a very recent paper, which was published in Nature Comm, maybe even after the current manuscript was submitted to the same journal (Lan et al, Nat Comm 15, 2544 [2024]). This latest paper together with the references listed by the authors in the manuscript show that the investigation of electron solvation dynamics and CTTS formation in general has become a research topic of great relevance within the physical chemistry science community.

In this context, the present manuscript adds the significant contribution of investigating the very first step involved in the metal-to-solvent electron transfer process. In fact, while the formation of CTTS and their consequent relaxation that brings to the solvated electrons final state has been studied and characterized extensively, little has been reported about the initial excitation and delocalization process. For all these reasons, I believe that this work deserves publication in Nature Communications.

Reply: We thank the reviewer for his/her positive comments. We added the Lan et al. paper into the Introduction section as Ref. 11 and we pointed out the differences in the processes observed in the core and valence regions. While the formation of the CTTS states through excitation/ionization at the core level is associated with the presence of the internal core-hole clock, there is no analogue in the valence-orbital region where the CTTS states are typically studied. We also place more emphasis on looking at the initial electronic state of the CTTS formation. We also changed the title to reflect the main message of the manuscript.

However, before giving my full recommendation, I would like the authors to comment on the following points:

Reviewer Comment 1.1 — My main concern is about the simple formula reported at the end of page 15 used to extract the wavepacket evolution time constant t_{WPE} from the electron spectra. I’m not sure whether I miss something, but I believe that the formula is not consistent with the observations and the plots reported in Fig. 5. In fact, as it is currently written, t_{WPE} increases the larger is the ratio between the intensities of the $2p^{-2}$ (from delocalized states) and $2p^{-2}3p$ peak (from localized states). From Fig. 3, it is apparent that for all the photon energies, the intensity of $2p^{-2}$ peak (not dispersive) is always larger than the $2p^{-2}3p$ peak (dispersive), and this is true for all the samples investigated. Consequently, the formula would return a value of t_{WPE} "larger" than t_{1s} basically for all photon energies, since the ratio is evidently always larger than one. Additionally, also the photon energy trend shown in Fig. 5 seems not consistent with the formula for t_{WPE} . Let’s have a look more in details at Fig. 3 for the Na+ case: at 1075 eV (below 1s threshold), from the 2D plot the $2p^{-2}$ peak is "relatively weak". There is not a color scale, but the maximum intensity of the peak is for photon energies above the 1s threshold, as it should be. On the other hand, the $2p^{-2}3p$ peak is faint, but its intensity can be considered comparable, at this photon energy, to the one of the $2p^{-2}$ peak. On the contrary, at 1080 eV (above the 1s threshold), the intensity of the $2p^{-2}$ peak is "large" and the one of $2p^{-2}3p$ peak is very small, making the ratio in the formula a very big number. As such, one would calculate that $t_{\text{WPE}_{1075\text{eV}}} \gg t_{\text{WPE}_{1080\text{eV}}}$ but Fig. 5 shows the opposite. The trend obtained from the formula would be also contrary to the intuitive picture that higher energy induces a shorter electron delocalization time. I would like the authors comment on these observations. Maybe to make things clearer, the authors could show linecuts of the 2D plots at two specific photon energies (for instance 1075 and 1080 eV for Na+ mentioned above) so that the readers can have an idea about the respective integrated peak intensities?

Reply: We thank the reviewer for discovering the error. In fact, the correct formula, consistent with the observation and the discussion in the text, is: $\tau_{\text{delocalization}} = \tau_{1s} \frac{2p^{-2}3p}{2p^{-2}}$, as the reviewer correctly noted. We corrected the formula in the manuscript on page 5 and page 17 and we apologize for the error.

Reviewer Comment 1.2 — Regarding the introduction, can the authors elaborate more in details why "indirect" techniques like core-hole-clock methods are advantageous with respect to "direct" ultrafast spectroscopy performed with attosecond pulses when it comes to investigate liquid samples? In the introduction, it is stated that the requirements are high rep-rate, low pulse energy, attosecond pulses. My understanding is that these are exactly the parameters matched by High order Harmonic Generation (HHG) sources. Admittedly the latter are not easily providing up to the soft X-ray photons, but in the literature there are a few examples of sources able to do that. Otherwise, soft X-rays FELs represent also a valid alternative.

Reply: The core-hole clock can be considered as a more traditional approach to electron dynamics compared to the present-day impressive development in the HHG technology or X-ray free-electron lasers. Note, however, that it is not presently possible to initiate the core excitation and follow the dynamics at the sub-fs timescale, especially in the liquid phase. The core-hole-clock technique was typically applied for molecules placed on metallic surfaces. The application of this technique for problems of aqueous

chemistry is so far rare. We point out the limitations of the direct methods in the Introduction and Conclusions sections.

Reviewer Comment 1.3 — If possible, it would be nice to have an exemplary figure representing the extraction of the information as presented in the text at the end of page 6 / beginning of page 7. Something like a line cut of the 2D plot with the PCI and Voigt curves used to fit the delocalized ($2p^{-2}3p$) and localized ($2p^{-2}$) states peaks.

Reply: We agree with the reviewer, a similar point was raised by reviewer 2. We present the detailed extraction of the data and the fitting of the data along with the intensity ratios in the SI.

Reviewer Comment 1.4 — I appreciated the presentation of Fig.2, showing the molecular orbitals and identifying where are the unoccupied states that would most likely host the delocalized electrons. But it is said that those orbitals have been calculated for the most intense transition "below" the 1s ionization threshold. Would this transition be in correspondence with one of the peaks seen in Fig 1b? If this is the case, these peaks are somehow visible for the small cluster (with 6 H₂O molecules, green solid line) but not so much for the large one (with 18 H₂O molecules, black solid line), for which the calculations show a broad spectrum without narrow peak features, in particular for Mg²⁺ and Al³⁺ ions. How is the peak identified then? Also for Al³⁺ case, apparently, the most intense peak for the small cluster is at a photon energy a few eV lower than the most intense 'peak' of large cluster. Is possibly this the reason for the quite different electron density maps shown in Fig. 2c?

Reply: A similar point was raised by all reviewers. We decided to significantly change the presentation of the data showing theoretical calculations of the core-excited states. (i) We relocated some of the theoretical methods to the SI and significantly expanded the text to include all details. (ii) We added a new core-excited-states analysis to include information about how much of the electron density in each individual excited state is localized on the metal cation and how much of the electron density is localized in the water environment. (iii) We modified Figure 2 (former Figure 1) – it now shows an analysis of the spectra for a given structure, with the natural transition orbitals corresponding to the most intense transitions. In addition, we added calculated spectra for 50 structures showing the contribution to the overall spectra attributable to the transitions localized on the metal cation for the smaller and larger cluster sizes [see Figure 2, panels (c) and (d)]. (iv) We changed the broadening factor for the solvated cations to reflect the lifetime broadening. We used 0.28 eV for Na⁺, 0.43 eV for Mg²⁺, and 0.44 eV for Al³⁺. (iv) We extended the discussion about the character of the excited states on pages 11-13 and in the SI.

The new presentation of the data could also answer the reviewer's comments by showing which particular transition corresponds to the natural transition orbital. In the case of Al³⁺, the higher broadening factor leads to a broadening of the first absorption peaks and corresponds better to the spectra for the larger cluster. The new data also show the differences in the character of the excited states between the three cations – while for Na⁺ and Mg²⁺ the onset of the spectra is associated with electron-density transitions located primarily on the metal cation, in Al³⁺ a significant part of the density lies in the water environment.

Reviewer Comment 1.5 — For completeness, since it is mentioned in the text and reported in Table S1, I suggest adding the $2p^{-2}3s$ dispersive line to Fig. 3 as well. Furthermore, these states

seem to show resonant enhancement above the 1s ionization threshold even beyond the $2p^{-2}3p$ states. Can the same analysis on the $2p^{-2}3p$ be performed also on the $2p^{-2}3s$ states? Would the conclusions be the same? Probably the qualitative trend (delocalization time getting smaller for higher energies) yes, but quantitatively also?

Reply: The weak dispersive lines were added to Figure 3. However, in the energy range we considered (just below and above the 1s ionization threshold), compared to the $3p$ feature, the $3s$ signal has much lower intensity and doesn't persist above threshold to the same degree.

Reviewer Comment 1.6 — A last final comment about data treatment and methodology: how was the off-resonant photoemission contribution subtracted by the integrated intensity of the localized states?

Reply: The experimentally determined intensity was assumed to consist of a direct contribution due to non-resonant photoemission and a resonant contribution due to spectator Auger–Meitner decay. The intensity of the direct contribution was estimated by determining the intensity of the spectral feature well below and well above threshold. The intensity well above the threshold is slightly lower than that well below the threshold due to a gradual reduction of the photoionization cross section. The intensity of the direct contribution was assumed to vary linearly between these two extreme points, and the corresponding contribution was subtracted from the experimental spectra.

Minor remarks:

Reviewer Comment 1.7 — Typo in Table S1: Na+ is indicated with double charge (2+) as Mg.

Reply: Thank you for discovering the typo. The text was corrected.

Reviewer Comment 1.8 — Fig S5: I think the caption does not reflect what is shown in the figure, there are no plots of the residuals from the fit, neither the green baseline is visible in any of the panels (only the red data and the blue fits are shown).

Reply: We thank the reviewer for discovering the error. The figure was updated.

Reviewer 2

E. Muchova et al. studied the electron transfer mechanism that is important to chemical reactions in aqueous solutions. This important process involves a core-level excitation of the 1s electron from metallic ion to the intermediated electronic states followed by the electron transfer to the delocalized neighboring solvent water molecules. The lifetime of the charge transfer of valence electron is estimated using so-called core-hole clock measurements associated with the absorption width and kinetic energy of Auger–Meitner (AM) electrons. The group has extensive expertise and knowledge on this field and the manuscript is submitted to Nature Communication for a broader readership. From my point of view, this is a good study in general. There are several aspects that require clarification before publishing in Nat. Comm. General remarks: Authors should carefully address which electron(s) is described since the dynamics involved in core and valence electrons at the same time. In this manuscript, the dynamics terms, for example, electronic wavepacket

and charge transfer are used for valence electrons, and the AM process and core-hole clock are for core electrons. These are used to describe the processes of dynamics. The “state” is usually used to describe the electronic configuration as a whole (including metal ion and water). Throughout the whole manuscript, I can tell that Authors mixing these all together and this may be a very confusing explanation for a broader readership. All the detailed comments can be found in the attached PDF document.

Reply: We would like to thank the reviewer for the generally positive assessment of our work and also for the suggestions that help to clarify the text. We have tried to modify the text according to the comments and suggestions to convey the message, even if the narrative is complicated due to the different electrons that appear in the processes studied. We would like to acknowledge the reviewer’s careful reading and accurate comments that have helped to improve the text immensely.

Reviewer Comment 2.1 — Minor suggestions are shown in the attached PDF document.

Reply: We thank the reviewer for the detailed comments in the manuscript. We corrected the text accordingly.

Reviewer Comment 2.2 — Although you provided the lifetime of the core-excited metallic ion in water, the most attractive part of your study is that you are able to determine the core-excited CTTS of metallic ion in aqueous solution. From your title, it is hard to link any attracting feature that directly related to your study. You may need to consider a better title.

Reply: We thank you for the suggestion. We changed the title to make it more informative.

Reviewer Comment 2.3 — All the orbitals should be in italic form, for example, 1s should be *1s*.

Reply: We consistently corrected the symbols throughout the text.

Reviewer Comment 2.4 — Actually, I believe the importance to Authors’ studies is in the field of the Radiolysis of liquid water (similar to ref 34 that brings up the related filed of radiochemistry) and associated charge transfer dynamics, especially with ions in water irradiated by the high energy X-ray. This study here actually provide very valuable information of the early stage of radiolysis of aqueous solution containing different metal ions, which may eventually lead to the corrosion of the nuclear power plant and container of nuclear waste etc. If abstract and introduction can have some touch on this aspect, it will raise more attention and impact to this publication. This may eventually require you to change your title accordingly.

Reply: In fact, the early stages of radiolysis were poorly understood in the golden age of radiation chemistry in the 1960s. Current experiments, either time-resolved or static, with tunable X-rays will most likely enable a reinterpretation of the generally accepted knowledge in this field. As an outlook, we added a note in the Conclusions section. However, the central focus of this work is on the electron dynamics and we fear that opening a new narrative line would be confusing for the reader.

Reviewer Comment 2.5 — In INTRODUCTION and reference 2-11, these are charge-transfer from the valence excitations, and it is different from your study which involves in the core-excitation

process (followed by CTTS). It will be clear for readers to understand your study with a diagram to point out how you excite target system and which electron is transfer and how this is different from the charge transfer from valence excitation.

Reply: We added Figure 1 in the Introduction section to clarify not only the core-hole-clock method, but also which electrons are excited and ionized. We hope that the new figure and text restructuring will help the reader to better understand the processes discussed.

Reviewer Comment 2.6 — Please also cite this paper: Science 383, 1118–1122 (2024); this is the first Attosecond X-ray pump and attosecond X-ray probe on liquid water. (using XFEL). It will be after your Ref 16.

Reply: We added the proposed paper as Ref. 18.

Reviewer Comment 2.7 — In “EXPERIMENTAL SETUP” section, Authors explain the 2D, however, there is no figure to help description of the details. It will be helpful to readers that Figure 3 is moved to the front of the manuscript and incorporated with the current Figure 1 (since it is a partial or full projection of the current Figure 3). The detailed data fitting associated with AM peaks (PCI, resonant and non-resonant) should be presented in SI. These are missing.

Reply: We agree with the reviewer and we have discussed moving the 2D maps. In the end, we decided to expand the Experimental Methods section and to add a detailed data analysis in the SI and not to move the 2D maps. We believe that this presentation of the data will be clear for the readers.

Reviewer Comment 2.8 — “THEORETICAL METHODS” should be moved to SI and only a few sentences are needed to summarize the experimental support by calculations.

Reply: We reflect the comment by moving a large part of the Theoretical Methods section to the SI and adding details that were requested by reviewer 3. The SI now contains all details, derivations, and equations, only summaries of the methods and references are provided in the manuscript.

Reviewer Comment 2.9 — In “RESULTS AND DISCUSSIONS” section, the description in “X-ray absorption. We start with . . . , thereby reflecting the electron dynamics” seems to work better in INTRODUCTION for readers. This part is highlighted yellow in the attached PDF.

Reply: We agree and moved the text as suggested.

Reviewer Comment 2.10 — I have a trouble understanding the “final decay states” though out the manuscript. Do you mean it is the final state of AM process? Or even this state will decay to different state(s)? Please clarify this.

Reply: By this we mean the final states after the AM process, we clarified it on page 6. We changed the text accordingly to avoid confusion.

Reviewer Comment 2.11 — In the caption of Figure 1, there is no explanation of 0.2 eV phenomenological value. Why 0.2 eV? And justification?

Reply: The broadening value is kind of arbitrary, we usually used 0.2 eV in our previous calculations, however, the reviewer is right. The broadening accounts for (i) thermal and vibrational broadening, (ii) inhomogeneous broadening with a second solvation shell, (iii) limited statistical sampling, and (iv) finite lifetime of the formed state. We modified Figure 2 in the manuscript and used the broadening factors that reflect the lifetime broadening. We discuss the values on page 9.

Reviewer Comment 2.12 — The theory spectral shown in Figure 1(b) seems to have no help to Figure 1(a). I cannot see any resemblance between the data and calculations. And why only 6 and 18 waters are used to calculate spectra? This needs to be clarified.

Reply: For solvated cations, it has been reported that the first solvation shell contains 6 water molecules and the second solvation shell contains 18 water molecules. In our calculations, we thus model the first and second solvations shells, we emphasise this point on pages 9-13. For further discussion, we will focus mainly on the onset of the spectra. We modified the discussion and Figure 2, clarified the text, and added an explanation to the SI because a similar point was raised by all reviewers.

Reviewer Comment 2.13 — All the “panel (b) of Figure 1” should be changed to “Figure 1(b)” if possible.

Reply: The figure was updated.

Reviewer Comment 2.14 — The legend boxes in Figure 1 overlap with figure frames in 1(a) Mg²⁺ and 1(b)Al³⁺. In addition, the X axis only has 3 minor ticks for 5 eV span. This is not easy to locate energy positions.

Reply: The figure was updated and we made sure that the legend boxes do not overlap with the figure frames.

Reviewer Comment 2.15 — Figure 2: need to specify your metal color in the caption.

Reply: The figure was modified, the metal color is now clear.

Reviewer Comment 2.16 — Please clarify what the electron size is (physical picture)? I know you have that in equation shown in Theory part, but it is not clear in terms of physical picture of this parameter.

Reply: We expanded significantly the discussion in the SI on the character of the excited states and also addressed the relationship between the localized/delocalized character of the excited state and the electron size.

Reviewer Comment 2.17 — Please mark 2p-23s line in the current Figure 3.

Reply: The figure was updated. Reviewer 1 in comment 1.5 asked for the same correction.

Reviewer Comment 2.18 — The axes and ticks for Figure 3 are very hard to read. Please change it to a thicker axes and ticks, and maybe consider different color that is different from the dark 2D map.

Reply: Figure 3 was updated accordingly.

Reviewer Comment 2.19 — The “contributions” shown on the 4th line in “Electron delocalization dynamics” is not clearly specified. In addition, I can not understand why at this moment Authors start to discuss how core-hole clock works. This should be moved to introduction section.

Reply: We agree with the reviewer. We moved a part of the text to the Introduction section to clarify the core-hole-clock method. We expanded the discussion of spectator and normal Auger-Meitner contributions to the signal in the SI.

Reviewer Comment 2.20 — The parameter “ Γ ” is not defined at the beginning in the manuscript. M is not defined either.

Reply: The “ Γ ” parameter is now appropriately defined in the Results section, M is defined at the beginning of the Theoretical Methods section.

Reviewer Comment 2.21 — The intensity ratio between $2p^{-2}3p$ and $2p^{-2}$ peaks should be presented in the manuscript (with peaks shown Figures) and details analysis should be in SI. These are lacking.

Reply: We agree and we added the intensity-ratio information to the SI section. The same point was raised by reviewer 1 in comment 1.3.

Reviewer Comment 2.22 — The time constant τ WPE is not defined at the beginning of its appearance. Does WPE represent “wavepacket evolution”? I have to guess it.

Reply: We define the time constant as delocalization time in the Introduction section and use this term consistently throughout the manuscript.

Reviewer Comment 2.23 — Authors used ionization thresholds for IE and BE throughout the manuscript (figures). Please be consistent.

Reply: We modified the manuscript accordingly and decided to retain both ionization energy (IE) and ionization thresholds because these terms are not confusing.

Reviewer Comment 2.24 — What kind of tunneling do you mean on page 17? The electron tunneling during the CTTS? I cannot understand how your Figure 2 supports Figure 5 on page 17. There is no explanation of the colored shaded area following the data lines. Error bars? If so, what confidence levels?

Reply: We corrected the text on former page 17, in our case we mean tunneling of the wave function through an energy barrier. Data in Figure 5 were obtained from the 2D plots presented in Figure 2 (now Figure 3), the details are now discussed in the SI.

Reviewer Comment 2.25 — It is striking to me that it has longer CTTS time scales when water molecules are closer to metal ions. Can you clarify it? Is that because it contains more metallic character when waters bond to metal ions and those bonding does not speed up the electron transfer

to the outer free water molecules? This means that the ratio between $2p^{-2}3p$ and $2p^{-2}$ peaks will be larger for Al³⁺ case?

Reply: The delocalization timescales are longer for Al³⁺ because the electron is closer to the metal cation (Al³⁺ has the highest charge, resulting in a more organized and closed solvation shell) and the character is only partially localized on the metal cation.

Reviewer Comment 2.26 — A figure will be needed to explain why the speed up transfer of electron for all ions in water shown in Figure 5 and on page 18. How does the empty water states overlap with your excited water wavepacket (in a figure). This is an important information from your studies.

Reply: The reviewer is right that varying delocalization times for the three cations are an important information of the study and that the interpretation in the text is qualitative. After a long discussion, we decided not to aim at providing a quantitative overlap between the empty states of liquid solutions and excited-state wave packets because of the complexity of determining the solution-phase energy levels (<https://doi.org/10.1103/PhysRevB.76.235406>). We modified the discussion in the Conclusions section.

Reviewer Comment 2.27 — This statement is intuitive and good to let reader know it. I was wondering if you are able to have calculation to support the observed time constant or trends? So far, there is no theory support on your observation time scale as a function of photon energy, right?

Reply: Unfortunately, we cannot support the observed trends with calculations as this currently represents a real challenge to the theory. We discuss this point in the Conclusions section.

Reviewer Comment 2.28 — Can you also provide an outlook on: how the attosecond pump and attosecond probe at FEL help to verify your measurements?

Reply: We provide an outlook in the Conclusions section.

Reviewer 3

The authors report an interesting study coupling attosecond pump probe measurements with static quantum chemical calculations to evaluate the charge transfer time of mono-, di- and tri- charged ions in aqueous solution after inner layer ionization (1s shell). They measure the rate of wave packet delocalization in less than a femtosecond. This work sheds light on the stages preceding electron solvation. I find the writing of the manuscript not very didactic and complicated to follow for a non-specialist. It seems to me that proofreading could help improve this point. I'm not an expert on the experimental part, which I'm not in a position to criticize outright. However, my reservations about the current version of the manuscript (see below) lead me to recommend resubmission with major modifications.

Reply: We thank the reviewer for carefully reading the manuscript. We tried to make the text clearer and also include the comments from reviewers 1 and 2 to convey the main message.

Reviewer Comment 3.1 — The discussion of Figure 2 on page 12 is not very clear from my point of view. I find it hard to be convinced by the representation of the 3 NTOs in the Figure of the presence of a cage around Al^{3+} . Moreover, this analysis is based on a single conformation, right ? As the authors did calculations on 200 geometries, it would be useful to capture this diversity in the plot ?

Reply: We agree with the reviewer; all reviewers have raised the same point. We decided to significantly change the presentation of the data showing theoretical calculations of the core-excited states. (i) We relocated some of the theoretical methods to the SI and significantly expanded the text to include all details. (ii) We added a new core-excited-states analysis to include information about how much of the electron density in each individual excited state is localized on the metal cation and how much of the electron density is localized in the water environment. (iii) We modified Figure 2 (former Figure 1) – it now shows an analysis of the spectra for a given structure, with the natural transition orbitals corresponding to the most intense transitions. In addition, we added calculated spectra for 50 structures showing the contribution to the overall spectra attributable to the transitions localized on the metal cation for the smaller and larger cluster sizes [see Figure 2, panels (c) and (d)]. (iv) We changed the broadening factor for the solvated cations to reflect the lifetime broadening. We used 0.28 eV for Na^+ , 0.43 eV for Mg^{2+} , and 0.44 eV for Al^{3+} . (iv) We extended the discussion about the character of the excited states on pages 11-13 and in the SI.

Reviewer Comment 3.2 — I'm surprised that the photoelectron stays in the vicinity of the cation long enough. I would expect it to escape further away ? A dynamical description is somehow missing to interpret fully the data. I agree that simulating the process delocalisation in real time is a challenge for theory, but this is where simulations would make sense. Shouldn't this challenge be taken up for an article in Nat. Commun? Considering fixed core positions (static calculations) may be a problem, even on such short time scales. The remarkable H_2O^+ simulations published by several groups in recent years (notably the Czech school) have shown a close coupling between core dynamics and the degree of gap delocalization. This makes me wonder about the present case.

Reply: It is indeed one of the surprising results of our experiment: on the attosecond timescale, there is no qualitative difference between the outgoing photoelectron and the electron residing in the bound excited state. The reviewer is correct that the time-dependent image would provide information about the initial stages of the electron delocalization, but the simulations are not currently feasible. The mentioned nuclear-dynamics studies on H_2O^+ were not carried out for ultrafast decaying core-excited states of solvated metals formed by the environment, but for water dimers in the gas phase. In the present case, we had to deal with the electron dynamics in the core-excited state before the nuclei start moving, and this has not yet been successfully solved theoretically.

Reviewer Comment 3.3 — The theoretical part lacks a great deal of technical detail to enable us to pinpoint the type of simulation carried out and the robustness of the results obtained. I'm not claiming that the calculations were poorly done, but referring to reference 34 without explanation isn't enough. The same applies to the absorption X. These details could be placed in SI. They should enable a third-party researcher to reproduce the article's results.

Reply: We expanded the Theoretical Methods section and moved it to the SI. We added details discussed in former Ref. 34 (now Ref. 36) so everyone has the full potential to reproduce the results. Since reviewer 2 requested to shorten the Methods section, we chose to add all details to the SI.

Reviewer Comment 3.4 — Furthermore, which simulation results are taken from reference 34 and which are new to the present article? This is not made explicit in the manuscript.

Reply: Ref. 34 (now Ref. 36) is the joint work on the interpretation of intermolecular Coulombic decay (ICD) in the solutions of the simple cations Na^+ , Mg^{2+} , and Al^{3+} . To this end, we performed classical large-box molecular dynamics simulations with corresponding solutions to generate a sufficient number of representative structures and scalable models for further calculations. We used these structures for *ab initio* calculations in Ref. 34, as starting structures for very different types of *ab initio* calculations in our more recent paper Nat. Chem. **15**, 1408-1414 (2023) and as starting structures in the present work. These initial structures are the only result from Ref. 34. We clarified this point in the Theoretical Methods section.

The experimental data for this study were measured in the same beamtime as those for two other studies from the same author team, namely Ref. 34 (now Ref. 36) and Gopakumar et al., DOI: 10.1038/s41557-023-01302-1 (2023). However, in these previous works the focus was on non-local autoionization channels, and consequentially different kinetic-energy regions of the photoemission spectra were analyzed. To clarify this, we added a sentence at the end of the Experimental Methods section.

A core-level spectrum of Na shown in the Supplementary Material of Ref. 34 (now Ref. 36) to illustrate the binding-energy analysis of this work was now used to extract the lifetime broadening of the Na 1s level shown in Supplementary Figure S6 of this manuscript. To clarify this, we added a citation.

Apart from this single spectrum in the Supplement, all data presented in the manuscript are new and have not been presented in any previous work.

Reviewer Comment 3.5 — Page 3 : EUV is not defined

Reply: The abbreviation was defined on page 3.

REVIEWER COMMENTS

Reviewer #1 (Remarks to the Author):

I'm glad to see that the comments and suggestions of all the referees have been properly addressed and eventually taken into accounts by the authors. The text of the manuscript has been modified according to the referees' reports and in my opinion it is significantly improved.

I'm satisfied by the discussions they presented in their reply letter. As a consequence, my recommendation is to publish this work in Nature Communications in the present form.

Reviewer #2 (Remarks to the Author):

Muchová et. Al. resubmitted their revised manuscript titled "Attosecond formation of charge-transfer-to-solvent states of aqueous ions probed using the core-hole-clock technique". Authors have made significant changes to their manuscript. I still have some comments on their manuscripts and some of them are shown in the pdf in yellow remarks. Please see the attached PDF and word file.

Reviewer #3 (Remarks to the Author):

The authors have thoroughly revised their manuscript, taking most of my remarks into consideration, apart from the simulations of delocalization dynamics, which could have been tried much harder. This is the authors' choice. I therefore recommend publication of this manuscript.
Sincerely

Muchová et. Al. resubmitted their revised manuscript titled “Attosecond formation of charge-transfer-to-solvent states of aqueous ions probed using the core-hole-clock technique”. Authors have made significant changes to their manuscript. I still have some comments on their manuscripts and some of them are shown in the pdf in yellow remarks.

1. In Figure 1, this is a good figure. There are some comments that need to be addressed to make it clear for readers. Please see the remarks in PDF. For example:
 - a. Blue spectrum in (b) is not clear in its color.
 - b. Environment is not clearly specified: I guess this is your water molecules around the metal ions.
 - c. $2p$ and $3p$ orbitals are not labelled however the electron configurations are given.
2. Minor: the sentence “Furthermore, currently, the application of attosecond X-ray pulses to aqueous-phase photoemission spectroscopy is hampered by the low pulse repetition rates of the available light sources, forcing the use of overly high pulse intensities and leading to significant and difficult-to-quantify space-charge effects.”
 - a. I guess you meant the low repetition rate of the attosecond X-ray that gives low signal-to-noise level. In order to get sufficient signal, a high pulse energy was used and that inevitably leads to a strong space-charge effect? Do you have any references on this aspect?
 - b. As I know, LCLS-II is running at 8.3 kHz and soon to ~50 kHz and higher. I guess in the near future, the statement will not be valid.
3. Minor: what is the distance from the liquid jet to the entrance of electron analyzer? And what is the collection efficiency? I know it is usually low. Do you have some estimated numbers?
4. In your sentence: “Signal integration in the vertical direction, within the kinetic-energy range of the AM peaks, yields the respective partial-electron-yield X-ray absorption spectra, PEY-XAS.”
 - a. Isn't this an integration “horizontally” along the electron kinetic energy (Figure 3) so a PEY-XAS spectrum can be obtained?
 - b. This was pointed out in the last comments and Authors did not correct or reply to this comment.
5. For figure 2, it's good to have the comparison of 6 and 18 water clusters with metallic ions. Can you provide the area ratio for panels (c) and (d) so we can use that ratio to tell the local excitation property on the metallic ions? In addition, isn't

the 18 water case close to real case of what you have in experiments? If so, why shows 6 water? Does that have any help on your data interpretation? Or do you mean that your data will also have contribution of system that only contains 6 water clusters? It's very rare to have excitation of metal-water clusters only from first shell, right? By only considering the 18 water case, the core-excited states are all very diffused for all ions. If 6 water case is just an extent of your theory, it is better to remove it and to reduce the distraction (of course you can put it in SI).

6. Minor: not sure why your changed all "diagonal" features to "dispersive" features. To me, the diagonal features are easier to identify this signal. But if other reviewers ask you to do so, then keep it.
7. For Figure 4, a more systematic explanation of this figure should be made. In general, there are two processes at the beginning of the excitation (a) $1s^{-1}e$ (b) $1s^{-1}np_CTTS$ and both will evolve to $1s^{-1}e_{delocalized}$, right? The explanation you made are very confusing and jumping around between processes.
 - a. Maybe put the "wave-packet evolution" arrow on top of two color boxes (yellow and green) and put two "additional arrows" for $1s^{-1}e \rightarrow 1s^{-1}e_{delocalized}$ and $1s^{-1}np_CTTS \rightarrow 1s^{-1}e_{delocalized}$.
 - b. In addition, it will be very helpful to make this figure coherent with your Figure 1 to help readers in general. What I meant is to move this Figure earlier to be with Figure 1 (one figure with all these information) so we know how you do the lifetime calculations before we see your data. Once you lay down this groundwork, the data interpretation is straightforward.

Reviewer's 2 comments

Muchová et. Al. resubmitted their revised manuscript titled "Attosecond formation of charge-transfer-to-solvent states of aqueous ions probed using the core-hole-clock technique". Authors have made significant changes to their manuscript. I still have some comments on their manuscripts and some of them are shown in the pdf in yellow remarks.

We thank the reviewer for careful reading of the updated manuscript. We have included the majority of the suggested comments, the new text is highlighted in green. Point-by-point answers are provided below.

1. In Figure 1, this is a good figure. There are some comments that need to be addressed to make it clear for readers. Please see the remarks in PDF. For example:

a. Blue spectrum in (b) is not clear in its color.

The color was changed.

b. Environment is not clearly specified: I guess this is your water molecules around the metal ions.

The figure was updated to include the environment.

c. $2p$ and $3p$ orbitals are not labelled however the electron configurations are given.

The figure is a sketch, we used a typical notation in which we simply denote $1s$, valence, and virtual orbitals without any further specification.

2. Minor: the sentence "Furthermore, currently, the application of attosecond X-ray pulses to aqueous-phase photoemission spectroscopy is hampered by the low pulse repetition rates of the available light sources, forcing the use of overly high pulse intensities and leading to significant and difficult-to-quantify space-charge effects."

a. I guess you meant the low repetition rate of the attosecond X-ray that gives low signal-to-noise level. In order to get sufficient signal, a high pulse energy was used and that inevitably leads to a strong space-charge effect? Do you have any references on this aspect?

Yes, the referee has understood the meaning of the sentence: the applicable pulse energies are limited by space-charge effects, and the photoemission signals and signal-to-noise levels are correspondingly determined and limited by the repetition rate of the available light sources.

There is broad literature on space-charge effects in condensed-phase photoemission, e.g., G. Schönhense et al., *Rev. Sci. Instrum.* **92**, 053703 (2021). Such issues are prevalent in the aqueous phase too, for example, see R. Al-Obaidi et al., *New J. Phys.* **17**, 093016 (2015).

b. As I know, LCLS-II is running at 8.3 kHz and soon to ~50 kHz and higher. I guess in the near future, the statement will not be valid.

We agree that it will not be valid in the future, we refer to the current state.

3. Minor: what is the distance from the liquid jet to the entrance of electron analyzer? And what is the collection efficiency? I know it is usually low. Do you have some estimated numbers?

We added the details on page 7.

4. In your sentence: "Signal integration in the vertical direction, within the kinetic-energy range of the AM peaks, yields the respective partial-electron-yield X-ray absorption spectra, PEY-XAS."

a. Isn't this an integration "horizontally" along the electron kinetic energy (Figure 3) so a PEY-XAS spectrum can be obtained?

The reviewer is correct, we integrated the intensity in the horizontal direction, to obtain the PEY-XAS along the vertical coordinate. We corrected slightly the text on page 8. The assumption made here is that absorption is proportional to the Auger electron yield. The exact shape of the PEY-XAS spectrum can slightly depend on the particular Auger channel that is considered. This is discussed in large detail in R. Golnak et al., *Sci. Rep.* **6**, 24659 (2016).

b. This was pointed out in the last comments and Authors did not correct or reply to this comment.

We corrected the text to address the concerns.

5. For figure 2, it's good to have the comparison of 6 and 18 water clusters with metallic ions. Can you provide the area ratio for panels (c) and (d) so we can use that ratio to tell the local excitation property on the metallic ions? In addition, isn't the 18 water case close to real case of what you have in experiments? If so, why shows 6 water? Does that have any help on your data interpretation? Or do you mean that your data will also have contribution of system that only contains 6 water clusters? It's very rare to have excitation of metal-water clusters only from first shell, right? By only considering the 18 water case, the core-excited states are all very diffused for all ions. If 6 water case is just an extent of your theory, it is better to remove it and to reduce the distraction (of course you can put it in SI).

The reviewer is correct, adding the numbers improves the main point. The text was updated to include the information about the percentage of the surface area attributed to the local character of the excited states, data are now on page 13.

We would like to keep both cluster sizes in the text. The larger size might seem more realistic on the first glimpse; however, we want to include the results for both cluster sizes to also provide comparison to more accurate EOM-CCSD results and to previous results in Ref. 37.

6. Minor: not sure why your changed all "diagonal" features to "dispersive" features. To me, the diagonal features are easier to identify this signal. But if other reviewers ask you to do so, then keep it.

We changed the term to comply with the existing literature.

7. For Figure 4, a more systematic explanation of this figure should be made. In general, there are two processes at the beginning of the excitation (a) $1s-1e$ (b) $1s-1np_CTTS$ and both will evolve to $1s-1edelocalized$, right? The explanation you made are very confusing and jumping around between processes.

a. Maybe put the "wave-packet evolution" arrow on top of two color boxes (yellow and green) and put two "additional arrows" for $1s-1e \rightarrow 1s-1edelocalized$ and $1s-1np_CTTS \rightarrow 1s-1edelocalized$.

We thank the reviewer for the suggestions. We decided to put A, B, and C to better correlate the text and the figure. We also decided not to add more arrows since it seems to complicate the figure and make the graphics messy.

b. In addition, it will be very helpful to make this figure coherent with your Figure 1 to help readers in general. What I meant is to move this Figure earlier to be with Figure 1 (one figure with all these

information) so we know how you do the lifetime calculations before we see your data. Once you lay down this groundwork, the data interpretation is straightforward.

We understand the point raised by the reviewer, but after a discussion among the authors we think that this change would not lead to the clarification of the text, so we would like to keep Fig. 4 in its place.

Additional comments from the pdf:

p.4 Maybe here you can clearly point out that the CTTS (with electron from the core-excitation) is the state that relaxes in parallel with the core-hole relaxation?

We kept the text as it was, the formation of CTTS states is discussed in the next paragraph.

p. 5 can you add: (below the ionization energies of metal ions); it will be also very helpful to mention that these virtual orbitals has strong interactions with water molecules to form the CTTS.

The suggested text was added.

I believe you refer this to the spectrum in (b)?

The text was corrected.

p. 8 and the larger cluster includes the hydration shell upto the 2nd layer??

The text was corrected.

p. 8 how is this work relating to your CTTS here? It comes out of nowhere in this experimental section. Will this be better in the introduction?

The phrase was added because of the concern of Reviewer 3 to state explicitly which data are taken from the previous work.

p. 8 and which are new to the present article? This is not made explicit in the manuscript. I agree that integration of the 2D map along the vertical direction gives you electron spectra but this should not be called PEY-XAS, right? This is confusing and please verify it.

PEY-XAS is the correct term (in fact, integration proceeds along the horizontal direction, as was correctly annotated by the reviewer earlier). The assumption made here is that absorption is proportional to the Auger electron yield. The exact shape of the PEY-XAS spectrum can slightly depend on the particular Auger channel that is considered. This is discussed in large detail in R. Golnak et al., Sci. Rep. **6**, 24659 (2016).

p. 10 in current case: from metal ion to water molecule?

The text was corrected.

p. 10 what is your excited-state eigenstates here? This is very confusing.

On p. 10 we discuss the process from the time-dependent perspective. However, as we mention in the theoretical section, we performed calculations within the time-independent perspective and as we

also elaborate further in the text, the eigenstates within the time-independent perspective are not fully formed within the timescale of the core-hole lifetime.

p. 10 what do you mean by electronic wave packet in the core-hole state?, From your definition, your wavepacket is specifically related to the excited electron and how does this electron in the core-hole state? Did you mean that this wavepacket is associated with the core-hole and the decay of the core-hole (AM) will affect the wavepacket evolution?

We mean the electronic wave packet of the core-hole state, the text was modified.

p. 11 and interaction from the surrounding water molecules?

We did not modify the text.

p. 12 maybe put a short sentence to describe that NTOs contain partial np characters from metal and delocalized water molecules.??? as you did in the main text.

We did not modify the figure caption; the local np character is pointed out in the text for panel (b).

p. 13 can you use a very short sentence to describe σ_e (electron size). From what I read below, it is an average distance of the excited electron away from the metal center?

The text was modified.

p. 16 what is the state before this later AM decay? In this figure, the horizontal axis is the time scale evolution and how do you get this $1s-1e_{\text{delocalized}}$? Is this from direct ionization or resonant excitation (or both)? This needs to be clarified. From what I read below, it can be from both but in your explanation, you linked to CTTS state. Later you linked to $1s-1e$ as well. So basically it ($1s-1e_{\text{delocalized}}$) can be from both.

We added letters A, B, and C to clarify the states / processes we are describing. Figure 4 is complex, the $1s-1e_{\text{delocalized}}$ states can be formed after both excitation and ionization, which we claim in the caption.

p. 17 what is M for here? or do you mean AM?

The text was corrected, by M we mean a metal.

p. 17 how do you remove it?

The non-resonant contribution was estimated by taking the average of the $2p^23p$ peak area for the five lowest photon energies in each map. We added a sentence to the caption of Table S2 in the SI to explain the procedure.

p. 17 I don't think this sentence explains much on your observation. If the spatial extent and ion-water distance decrease, how come the lifetime is higher for Al^{3+} ?

We believe that it does, the formation of the tight cage around Al^{3+} connected to its higher charge in fact hampers the formation of the CTTS states.

REVIEWERS' COMMENTS

Reviewer #2 (Remarks to the Author):

Muchová et. Al. resubmitted their revised manuscript titled "Attosecond formation of charge-transfer to-solvent states of aqueous ions probed using the core-hole-clock technique". Authors have made significant changes to their manuscript and adequately addressed all the comments after three times of reviews. I am glad that they made all these changes. No further comments from my side. This manuscript is ready for the publication at Nature Communications.